# `BEDTime`: A Unified Benchmark for Automatically Describing Time Series

**Medhasweta Sen** [1]   **Zachary Gottesman** [1]   **Jiaxing Qiu** [1]   **C. Bayan Bruss** [2]   **Nam H Nguyen** [2]   **Thomas Hartvigsen** [1]

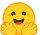 https://huggingface.co/datasets/HartvigsenGroup/BEDTime

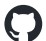 https://github.com/hartvigsen-group/bedtime

## Abstract

Recent works propose complex multi-modal models that handle both time series and language, ultimately claiming high performance on complex tasks like time series reasoning and cross-modal question answering. However, they skip foundational evaluations that such complex models should have mastered. So we ask a simple question: *How well can recent models describe structural properties of time series?* To answer this, we propose that successful models should be able to *recognize*, *differentiate*, and *generate* descriptions of univariate time series. We then create **BEDTime**, a benchmark to assess these novel tasks, that comprises **five datasets** reformatted across **three modalities**. In evaluating **17 state-of-the-art models**, we find that (1) surprisingly, dedicated time series-language models fall short, despite being designed for similar tasks, (2) vision language models are quite capable, (3) language only methods perform worst, despite many lauding their potential, and (4) all approaches are clearly fragile to a range of real world robustness tests, indicating directions for future work. Together, our findings critique prior works' claims and provide avenues for advancing multi-modal time series modeling.[1]

## 1. Introduction

Time series are data repeatedly collected over time to measure how environments change. Decision-making in crucial domains, like medicine (Orayj, 2025; Lu et al., 2023) and finance (Hosseini et al., 2025; Ji et al., 2019), hinges on accurately analyzing time series data. However, the history of machine learning models for time series, regardless of task, has mostly been uni-modal (Wang et al., 2024; Tan et al., 2024; Benidis et al., 2022; Sezer et al., 2020). So many recent works have proposed multi-modal models that can "reason" about time series using language (Chow et al., 2024; Yu et al., 2023; Wang et al., 2025a; Xie et al., 2025) and solve new multi-modal tasks (Merrill et al., 2024; Tan et al., 2025; Xue & Salim, 2023; Jin et al., 2024). These approaches stand to unlock more-complex tasks, but make fair evaluations challenging, yet crucial.

Prior evaluations of language-based time series models have two key limitations. First, models are often proposed alongside specialized evaluation datasets and then evaluated in near-isolation, leaving a lack of head-to-head comparisons between models. Such comparisons are crucial to track progress. Second, dedicated benchmarking efforts (Merrill et al., 2024; Cai et al., 2024; Fons et al., 2024; Tan et al., 2025) often focus on complex reasoning tasks. However, complex behaviors should build on established, fundamental model capabilities. Evaluating simpler, more fundamental, tasks is therefore needed to better isolate and assess these models' capabilities and potential failure-modes.

We propose that describing key structural features (Zhang et al., 2025) in univariate time series data is a fundamental capability for more-complex time series reasoning tasks (Chow et al., 2024)–analogous to addition being foundational for math reasoning. Univariate time series have extensive real-world applications across domains like medical diagnostics (Aminorroaya et al., 2025), market prediction (Mehtab & Sen, 2020), and monitoring physical systems (Elsaraiti & Merabet, 2021). They also underpin widely-adopted benchmarks and datasets, spanning forecasting (Makridakis et al., 2018) and classification (Bagnall et al., 2017). Further, describing univariate time series with language is an important task on its own for both general-purpose (Trabelsi et al., 2025; Ito et al., 2025; Chow et al., 2024) and task-specific settings (Han et al., 2024; Martínez-Cruz et al., 2021; Serrano Chica, 2020; Martínez-Cruz et al., 2023). For example, Law et al. (2005) found that neonatal ICU staff picked better treatments when shown descriptions

---

[1]University of Virginia, Charlottesville, Virginia, USA [2]Capital One, USA. Correspondence to: Medhasweta Sen <jwm9fu@virginia.edu>, Thomas Hartvigsen <hartvigsen@virginia.edu>.

*Proceedings of the 43rd International Conference on Machine Learning*, Seoul, South Korea. PMLR 306, 2026. Copyright 2026 by the author(s).

[1]All of our code and data needed to reproduce our results is public.

of physiological time series, which has recently been corroborated in finance (Yarushkina et al., 2025). However, no evaluations test whether state-of-the-art multi-modal time series models can describe univariate time series.

To address these limitations, we introduce BEDTime, the first rigorous benchmark enabling direct, task-specific comparisons of models for recognizing, distinguishing, and generating general natural language descriptions of univariate time series. We propose that a successful multi-modal time series reasoning model should be able to perform at least three tasks: (1) *recognize* accurate descriptions when shown a corresponding time series, (2) *differentiate* correct descriptions from incorrect descriptions, and (3) *generate* language descriptions that capture key structural properties of an input time series. We create BEDTime by unifying five recent datasets, formatting them to support each of our proposed tasks and including evaluation metrics, resulting in 46,843 time series with corresponding descriptions.

We use BEDTime to extensively compare 17 Large Language Models (LLMs), Vision-Language Models (VLMs), and recent pre-trained time series language models (TSLMs) in a large set of head-to-head comparisons, spanning both automated metrics and human evaluations. Overall, we find that (1) VLMs are successful despite prior works rarely including them as comparisons, indicating they are underused in this area, (2) prompting LLMs is largely unsuccessful, despite popular works lauding text-only time series analysis, (3) pre-trained TSLMs consistently outperform comparably-sized LLMs, while lagging behind VLMs, and (4) all models are susceptible to realistic robustness tests. Together, these findings suggest additional baselines for future works and indicate many open problems for the community to address.

Our work contributes to the literature as follows:

1. We introduce BEDTime, the first benchmark for automatically describing time series. BEDTime unifies five datasets, formatted for three proposed tasks, across three modalities.
2. Our evaluation of 17 state-of-the-art models suggests that future works should prioritize explicit modeling of structural features of time series and that recent methods are generally fragile to realistic perturbations.
3. We introduce comprehensive evaluation methods, combining both human expert assessments and automated metrics. All relevant code and data has been made public.

## 2. Related Works

**Benchmarks for Multi-Modal Time Series Reasoning.** Recent benchmarks for time series reasoning fall into three categories: synthetic datasets, empirical human-annotated datasets, and domain-specific Question-Answering datasets. Synthetic datasets such as TRUCE-Synthetic (Jhamtani &

Berg-Kirkpatrick, 2021), TaxoSynth (Fons et al., 2024), and SUSHI (Kawaguchi et al., 2024) allow controlled evaluation over trends, periodicities and stochastic fluctuations but often lack the nuances of real world time series. TRUCE-Stock (Jhamtani & Berg-Kirkpatrick, 2021) is one of the few empirical datasets with real-world stock-data time series and crowd-authored descriptions. QA-style evaluations like TimeSeriesExam (Cai et al., 2024) assess reasoning via templates, while ECG-QA (Oh et al., 2023), DeepSQA (Xing et al., 2021), and PIXIU (Xie et al., 2023) target domain-specific question answering with time series as auxiliary input but often rely on machine-generated language that is not generalizable across domains or even across different datasets of the same domain. Moreover, existing benchmarks lack multimodal coverage, evaluation frameworks for open-ended description generation, and robustness testing under real-world perturbations; BEDTime includes all three, and will release all code and data to facilitate reproduction and evaluation of new datasets and models upon acceptance. Table 4 situates BEDTime among existing multi-modal time series and language benchmarks.

**Methods for Multi-Modal Time Series Reasoning.** Recent work on time series reasoning spans three paradigms: LLMs, VLMs, and recently TSLMs. (1) *LLMs:* LLM-TIME (Gruver et al., 2023) treats forecasting as next-token prediction; PromptCast (Xue & Salim, 2023) frames it as a question-answering set-up. Several surveys (Zhang et al., 2024; Jiang et al., 2024) outline prompting and fine-tuning strategies, but highlight LLMs' limited numeric reasoning. (2) *VLMs:* Models like CLIP (Radford et al., 2021), BLIP-2 (Li et al., 2023), and Time-VLM (Zhong et al., 2025) process plotted time series via vision-language alignment. These models fail to capture signal fidelity. (3) *TSLMs:* ChatTime (Wang et al., 2025a) introduces a foundation model that discretizes raw time series and encodes them into instruction-style prompts, enabling LLMs to perform time series question-answering with strong zero-shot generalization. ChatTS (Xie et al., 2025) leverages a synthetic data generator to align time series and text, by training a LLM to supports fine-grained reasoning and description generation over signals; (Trabelsi et al., 2025) uses cross-modal retrieval; CLaSP (Ito et al., 2025) learns joint embeddings; (Chow et al., 2024) integrate CoT with lightweight encoders. These methods all target alignment, captioning, or reasoning, but lack fundamental comparative evaluation. Table 5 summarizes current TSLMs and their capabilities. BEDTime unifies evaluation of LLMs, VLMs, and TSLMs across description tasks.

## 3. Method

As illustrated in Figure 1, we propose BEDTime, a benchmark for assessing models' ability to recognize, differen-

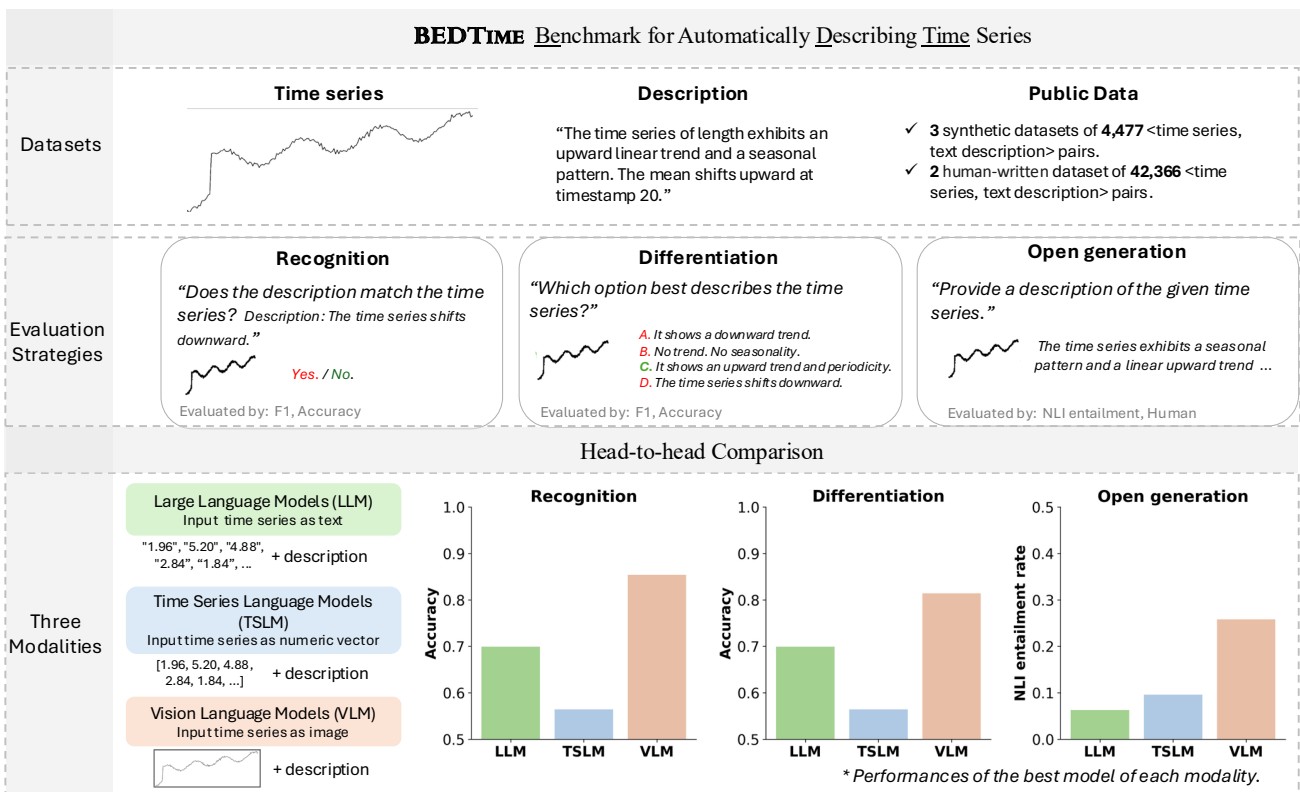

*Figure 1.* Overview of `BEDTime`, featuring head-to-head comparisons across three modalities. The benchmark includes two parts: a diverse collection of public datasets containing time series paired with textual descriptions of their structural properties (first row); and evaluation strategies across three tasks: description recognition, description differentiation, and open-ended generation given a time series (second row). The benchmark is compared head-to-head across three modalities (third row).

tiate, and generate descriptions of structural properties in univariate time series using natural language. `BEDTime` contains over 40,000 unique time series-descriptions pairs (90.44% real-world, 9.56% synthetic) , unified across five public datasets and formatted for evaluation across these three proposed tasks. Each task is fundamental to more complex and desirable time series understanding and reasoning capabilities (Chow et al., 2024; Cai et al., 2024). By focusing on simple, and interpretable descriptions of time series' structural properties like trend, periodicity, anomalies and abrupt changes, `BEDTime` enables evaluation of general-purpose language models without reliance on domain-specific knowledge, supporting broad cross-model comparison.

**Notation.** Consider a dataset $\mathcal{D} = \{(x_i, d_i)\}_{i=1}^N$ containing $N$ pairs of time series $x$ and corresponding descriptions $d$. Each time series $x_i \in \mathbb{R}^T$ is a real-valued sequence of length $T$, described by the corresponding natural language description $d_i$ as shown in Figure 1. A multi-modal time series foundation model $f(\cdot)$ may take in a time series $x_i$ and a prompt $p_i$ and produce an output $f(x_i, p_i)$ corresponding to the task description in $p_i$. Section 3.1 details how $p_i$ and $f(x_i, p_i)$ vary based on the task being evaluated.

### 3.1. Proposed Tasks

We propose three tasks that capable time series reasoning models should be able to perform. Tasks 1 and 2 are formulated as objective question answering, where models are a given time series and asked to recognize or differentiate correct and incorrect descriptions. Task 3 is a free-form response, where the models' describe the most salient structural properties of a given time series.

**Description Recognition (Task 1).** Given a time series and a description, a model must determine whether the description is consistent with the given time series. We format this as a True/False task and design the prompt $p_i$ to encourage $f(x_i, d_i) \in \{\text{“True”}, \text{“False”}\}$. Since our datasets do not inherently contain incorrect time series–description pairs, we construct negative examples by sampling within each dataset. For each time series, we use four distractor sampling methods to ensure robustness to the choice of negative sampling strategy. One method measures dissimilarity between descriptions, while the other three measure dissimilarity between time series and use the corresponding descriptions. In all cases, we select the most dissimilar description as the incorrect option. Appendix C provides further details on our negative sampling procedure.

**Description Differentiation (Task 2).** Given a time series $x_i$, the model must select the correct description $d_i$ from a set of four options $\{d_i, d_j, d_k, d_l\}$. This is posed as a multiple-choice task, with prompt $p_i$ formatted to present the options and elicit a letter-valued prediction $f(x_i, p_i) \in$ {"A", "B", "C", "D"}, in which each letter corresponds to a description option. A prediction is correct if it corresponds to the index of the true description $d_i$. As in Task 1, incorrect options are generated by sampling the three most dissimilar descriptions from the dataset.

**Open Generation (Task 3).** Given a time series $x_i$, the model must generate a natural language description $d_i = f(x_i, p_i)$ based on a prompt $p_i$ (e.g., "describe the given time series").

### 3.2. Dataset Description and Pre-processing

BEDTime comprises five public datasets, each reformatted for the tasks above: TRUCE-Stock (short, noisy real-world stock data with crowd-sourced descriptions) and TRUCE-Synthetic (short simple synthetic data with both crowd-sourced and template-based descriptions) (Jhamtani & Berg-Kirkpatrick, 2021), SUSHI (long, complex signals with prominent trends, seasonality, and noise and template-based descriptions) (Kawaguchi et al., 2024), TaxoSynth (variable-length taxonomic patterns based on critical aspects of frequently analyzed time series data with synthetic descriptions) (Fons et al., 2024), and NICU-HR (300 time-step real-world neonatal heart-rate signals with human-labeled descriptions grounded in clinical events) (Sullivan et al., 2024). These datasets are not originally designed for our specific tasks, so we perform several dataset-specific pre-processing steps. For TRUCE-Stock and TRUCE-Synthetic, when a time series has multiple valid descriptions, we create a separate entry for each unique series–description pair. For TaxoSynth, we use seven of the ten available datasets, focusing on univariate time series with associated qualitative descriptions. To be consistent with other datasets and recent benchmarks (Cai et al., 2024; Gruver et al., 2023), we remove all timestamps, keeping only the value sequences. For both the SUSHI and TaxoSynth datasets, we retain class and subclass labels, which were used to select incorrect options to ensure that distractor descriptions for recognition and differentiation tasks always come from different classes and subclasses than the target. For NICU-HR, we use the neonatal heart-rate recordings as fixed-length 300-step (2-second sampled) time series and their clinically validated natural-language time-series descriptions capturing variability and bradycardia events, and retain the 7-day mortality label to constrain distractor selection so that incorrect descriptions always come from patients in the opposite mortality class than the target.

## 4. Experiments

### 4.1. Experimental Setup

**Models.** We evaluate eight LLMs, six VLMs, and two TSLMs on BEDTime. This includes proprietary as well as open-weights models that range from 4.2B to 14B parameters.

**LLMs.** We evaluate GPT-5.4-mini (OpenAI, 2024), Gemini-3.0-flash (Google, 2025), GPT-4o (OpenAI et al., 2024), Gemini-2.0-Flash (Google, 2025), Llama-3.1-8B-Instruct (Grattafiori et al., 2024), Phi-3.5-Mini-Instruct (Abdin et al., 2024), Qwen2.5-7B-Instruct-1M, and Qwen2.5-14B-Instruct-1M (Yang et al., 2025). We convert time series to strings of comma-separated values, following prior works (Gruver et al., 2023; Cai et al., 2024). For GPT-4o, we also compare with the LLMTime (Gruver et al., 2023) prompting strategy, which we denote as LT.

**VLMs.** We evaluate GPT-5.4-mini-Vision (OpenAI, 2024), Gemini-3.0-flash-Vision (Google, 2025), GPT-4o-Vision (OpenAI et al., 2024), Gemini-2.0-Flash-Vision (Google, 2025), Qwen2.5-VL-7B-Instruct (Qwen et al., 2025), and Phi-3.5-Vision-Instruct (Abdin et al., 2024). Here, we represent the time series as matplotlib-rendered plots.

**TSLMs.** We evaluate ChatTS (Xie et al., 2025) and Chat-Time (Wang et al., 2025a), which operate on time series inputted directly as numerical vectors. [2]

**Metrics.** Since Recognition and Differentiation are objective question-answering tasks, evaluated on balanced datasets, we report accuracy. Since generation is harder to evaluate as the models generate free-form text descriptions, we use two approaches. We evaluate generation quality using both automatic and manual strategies. For automatic evaluation, we use a natural language inference (NLI) model and GPT-5 (OpenAI, 2025) to compute unidirectionally and bidirectionally entailment between generated and ground truth descriptions. The human evaluation criteria are adapted from prior literature on generating linguistic descriptions of time series (GLiDTS) work (Martínez-Cruz et al., 2023). Full evaluation details are provided in Section 4.2.2.

### 4.2. Results

#### 4.2.1. CAN FOUNDATION MODELS RECOGNIZE AND DIFFERENTIATE TIME SERIES DESCRIPTIONS?

We first compare all models on Recognition and Differentiation. For both tasks, to capture general trends, we also report

---

[2] We evaluate only these two TSLMs as they are currently the only TSLMs that support our tasks and provide publicly available code and pretrained weights. Table 5 summarizes other existing TSLMs and explains why they are incompatible.

*Table 1.* Overview of the five datasets in `BEDTime`, including the number of unique time series-description pairs, sequence lengths, and description characteristics and sources.

| Dataset | N | Series Length | Description Word Length | Description Source | Time Series Source |
|---|---|---|---|---|---|
| TRUCE-Stock | 5687 | 12 | 2–9 | Human | Google Finance API |
| TRUCE-Synthetic | 1677 | 12 | 1–8 | Human+Synthetic | Synthetic |
| TaxoSynth | 1400 | 24–150 | 5–40 | Synthetic | Synthetic |
| SUSHI | 1400 | 2048 | 5–60 | Synthetic | Synthetic |
| NICU-HR | 36679 | 300 | 8–60 | Human | NICU heart monitor |

*Table 2.* Accuracy of LLMs, VLMs, and TSLMs on recognition and differentiation tasks. LT denotes the LLMTime prompting strategy (Gruver et al., 2023). DTW distance is used to pick dissimilar descriptions. Reported ranks are intra-modal. For additional results, refer to Appendix E. *Note: TRUCE-Synth is short for TRUCE-Synthetic.*

| | Models | Datasets | | | | | Rank |
|---|---|---|---|---|---|---|---|
| | | TRUCE-Stock | TRUCE-Synth | SUSHI | TaxoSynth | NICU-HR | |
| LLMs | GPT-5.4-mini | **.68** | **.86** | **.89** | **.81** | **.63** | 1 |
| | Gemini-3.0 | .65 | .79 | .81 | .76 | .62 | 3 |
| | GPT-4o | .64 | .80 | .74 | .73 | **.57** | 3.6 |
| | GPT-4o+LT | .66 | .81 | .84 | .79 | .57 | 2.2 |
| | Gemini-2.0 | .60 | .72 | .72 | .69 | .56 | 5 |
| | Llama-3.1-8B | .50 | .54 | .49 | .51 | .46 | 7.8 |
| | Qwen2.5-14B | .61 | .72 | .51 | .59 | .54 | 5.4 |
| | Qwen2.5-7B | .65 | .63 | .51 | .57 | .53 | 5.6 |
| | Phi-3.5 | .59 | .52 | .54 | .57 | .51 | 7.2 |
| VLMs | GPT-5.4-mini | **.79** | **.88** | **.97** | **.9** | **.69** | 1 |
| | Gemini-3.0 | .73 | .84 | .91 | .87 | .66 | 2.6 |
| | GPT-4o | .75 | .83 | .96 | .88 | .60 | 2.4 |
| | Gemini-2.0 | .68 | .72 | .86 | .78 | .60 | 4 |
| | Qwen2.5-VL-7B | .61 | .72 | .61 | .73 | .54 | 5 |
| | Phi-3.5-vision | .69 | .66 | .79 | .68 | .53 | 5 |
| TSLMs | ChatTime-7B | .10 | .13 | .28 | .25 | **.54** | 1.8 |
| | ChatTS-14B | **.42** | **.60** | **.78** | **.77** | **.54** | 1 |

*(a)* Recognition

| | Models | Datasets | | | | | Rank |
|---|---|---|---|---|---|---|---|
| | | TRUCE-Stock | TRUCE-Synth | SUSHI | TaxoSynth | NICU-HR | |
| LLMs | GPT-5.4-mini | **.61** | **.84** | **.76** | **.76** | **.96** | 1 |
| | Gemini-3.0 | .58 | .75 | .72 | .68 | .95 | 2.6 |
| | GPT-4o | .52 | .73 | .65 | .71 | .94 | 3.8 |
| | GPT-4o+LT | .56 | .79 | .67 | .75 | .95 | 2.4 |
| | Gemini-2.0 | .53 | .63 | .63 | .61 | .94 | 4.6 |
| | Llama-3.1-8B | .26 | .28 | .23 | .26 | .69 | 8.4 |
| | Qwen2.5-14B | .53 | .68 | .45 | .53 | .81 | 5 |
| | Qwen2.5-7B | .43 | .47 | .24 | .35 | .78 | 6.8 |
| | Phi-3.5 | .32 | .37 | .44 | .50 | .76 | 7 |
| VLMs | GPT-5.4-mini | **.73** | **.88** | **.98** | **.84** | **.98** | 1 |
| | Gemini-3.0 | .68 | .83 | .94 | .77 | .96 | 2.4 |
| | GPT-4o | .67 | .81 | .97 | .81 | .96 | 2.4 |
| | Gemini-2.0 | .65 | .75 | .90 | .71 | .94 | 4 |
| | Qwen2.5-VL-7B | .62 | .73 | .87 | .76 | .79 | 4.6 |
| | Phi-3.5-vision | .61 | .71 | .76 | .70 | .77 | 5.8 |
| TSLMs | ChatTime-7B | .07 | .10 | .24 | .15 | .73 | 2 |
| | ChatTS-14B | **.41** | **.59** | **.60** | **.65** | **.82** | 1 |

*(b)* Differentiation

each model's intra-modal rank, averaged on all datasets. Our main results for both tasks are found in Table 2, which is broken down into Recognition (Table 2a) and Differentiation (Table 2b).

Overall, we observe that VLMs are consistently the best-performing models across each dataset and task, and GPT-5.4-mini is the best VLM. This is expected, as descriptions largely depend on structural properties, necessitating vision encoders. However, on noisier real-world data like TRUCE-Stock and NICU-HR, even the best VLM is surprisingly inaccurate, at these extremely simple tasks. We posit that a successful time series-text foundation model should near 100% accuracy for both description recognition and differentiation, so there is clear room for further method development on these tasks. LLMs are second best, with some of the best models rivaling VLMs in some cases (e.g., GPT-5.4-mini and GPT-4o on TRUCE-Synthetic and NICU-HR). GPT-4o also benefits from LLMTime prompting strategy, though only slightly. Notably, Qwen2.5-14B is the best open-weights model, suggesting a possible starting place for further development of such models. Increasing model scale (e.g., doubling LLM parameters from 7B to 14B) offers only marginal improvements. TSLMs show limited performance overall. While ChatTS outperforms

LLMs of comparable parameter size on TRUCE-Synthetic, SUSHI, TaxoSynth, and NICU-HR, it consistently does not match the performance of comparably sized VLMs and is also outperformed by larger LLMs and VLMs. ChatTime's performance is constrained by frequent refusals across all datasets except NICU-HR, resulting in below-random accuracy (see Appendix I for further analysis). While LLMs lag behind on long or structured sequences, well-trained TSLMs can match LLMs twice their size, particularly on datasets like SUSHI, TaxoSynth, and NICU-HR that reward temporal inductive biases. Nevertheless, VLMs still outperform TSLMs in the vast majority of settings, confirming that visual context remains a key strength across time series data input modalities. These trends hold for both Recognition and Differentiation task, as shown in Table 2a and Table 2b respectively, where the models are ranked similarly for both tasks.

Although VLMs outperform LLMs, their architectures still incorporate their base LLM components. To better isolate the contribution of their visual encoders, we compare each VLM directly with its LLM counterpart. Across datasets and tasks, VLMs irrespective sizes or model families consistently surpass their LLM baselines, irrespective of prompting strategies, underscoring the importance of visual rep-

*Table 3.* Evaluation results for time series description generation on SUSHI, showing (a) NLI entailment percentages for all models and (b) averaged human evaluation scores for the top model per modality. We assess the rate at which the LM description entails the ground truth (Gen → GT), the ground truth entails the LM description (GT → Gen), and when both entail each other (bi-directional entailment). For additional analysis and metrics, see Appendix J.

**(a) Automatic Evaluation (Natural language inference (NLI) entailment percentage (%))**

| | Models | Bi-directional | Gen → GT | GT → Gen |
|---|---|---|---|---|
| **LLMs** | GPT-5.4-mini | **5.43** | **35.21** | **8.86** |
| | Gemini-3.0 | 2.79 | 7.93 | 7.01 |
| | GPT-4o | 2.94 | 9.71 | 6.18 |
| | Gemini-2.0 | 2.21 | 15.43 | 2.43 |
| | Llama-3.1-8B | 0.07 | 1.00 | 0.07 |
| | Qwen2.5-14B | 1.93 | 7.21 | 2.07 |
| | Qwen2.5-7B | 1.50 | 6.21 | 1.79 |
| | Phi-3.5 | 1.21 | 1.57 | 1.50 |
| **VLMs** | GPT-5.4-mini | **16.29** | **58.00** | **31.14** |
| | Gemini-3.0 | 14.14 | 42.43 | 14.14 |
| | GPT-4o | 14.41 | 47.65 | 15.29 |
| | Gemini-2.0 | 2.79 | 8.00 | 5.14 |
| | Qwen2.5-VL-7B | 2.07 | 14.15 | 2.21 |
| | Phi-3.5-vision | 1.64 | 8.36 | 2.00 |
| **TSLMs** | ChatTime-7B | 1.08 | 2.21 | 1.29 |
| | ChatTS-14B | **2.65** | **23.53** | **2.65** |

**(b) Human Evaluation on 6 Designed Criteria**

| Model | Valid | Patterns | Position | Noise | Step-by-Step | Concise |
|---|---|---|---|---|---|---|
| GPT-4o-Vision | 1.00 | **0.891** | **0.751** | **0.815** | **0.853** | **0.851** |
| GPT-4o-Text | 1.00 | 0.361 | 0.234 | 0.624 | 0.681 | 0.715 |
| ChatTS-14B | 1.00 | 0.738 | 0.540 | 0.657 | 0.613 | 0.620 |

resentations for description recognition and differentiation, as shown in Figure 5. We also compute per-class accuracy on SUSHI, stratifying time series into 7 signal types of increasing complexity (Table 16). All modalities achieve near-perfect accuracy on simple classes (clean, smooth), but performance degrades with complexity, with LLMs most affected—smaller LLMs fall below random baseline on noisy signals—while VLMs remain robust across all complexity levels. TSLMs degrade more slowly than LLMs, confirming that numeric encoding provides robustness over text tokenization but not the full benefit of visual representation. Full details are in Appendix F. Appendix E shows consistent trends across distractor sampling strategies, confirming robustness.

### 4.2.2. CAN FOUNDATION MODELS GENERATE TIME SERIES DESCRIPTIONS?

We next evaluate the description generation capabilities of foundation models on the SUSHI dataset, which has long and richly-annotated descriptions, complex temporal structure (including trend, seasonality, and noise), and long sequence length (2048 steps). Given a time series, each model is prompted to generate a natural language description capped at 150 tokens. Example prompts are included in

Appendix D. We evaluate outputs using both automated and human-centered metrics. Results are summarized in Table 3 and discussed below.

**Automatic Evaluation.** We use a natural language inference (NLI) model (`deberta-base-long-nli`) (Sileo, 2024) to compute entailment between generated and ground truth descriptions for all models across all three modalities. Entailment is assessed bidirectionally (Machine Generated → Ground Truth and Ground Truth → Machine Generated), and we also report the proportion of strict mutual entailment. As shown in Table 3(a), GPT-5.4-mini achieves the highest entailment rates in both the LLM and VLM settings, with VLMs consistently outperforming their text-only counterparts. Among TSLMs, ChatTS-14B outperforms ChatTime-7B by a substantial margin. Across all modalities, entailment from generation to ground truth is consistently higher than the reverse, suggesting that models tend to produce more specific or detailed descriptions than those in the reference set.

**Human Evaluation.** We also conduct a human evaluation to assess whether generated descriptions are generic, accurate, interpretable, and fluent. Since human annotation is time-intensive, we evaluate a high-performing model per modality from Tasks 1 and 2: GPT-4o-Vision (VLM), GPT-4o-Text (LLM), and ChatTS-14B (TSLM). We selected 340 model-generated descriptions using stratified random sampling over the entire SUSHI dataset, ensuring that time series from all classes and subclasses were proportionally represented. Three annotators then scored these model-generated descriptions (40 overlapping for agreement, 100 unique per annotator), across three modalities, using **six** binary criteria: (1) *Coherence*: Descriptions must contain logically connected and meaningful statements. (2) *Pattern Identification*: Relevant features (e.g., trends, spikes, periodicity) are correctly recognized. (3) *Temporal Localization*: The positions of these features within the sequence are correctly specified. (4) *Noise Filtering*: Descriptions omit insignificant variations or noise. (5) *Abstraction*: Descriptions are high-level, avoiding overly detailed step-by-step repetition of the input sequence. (6) *Linguistic Quality*: Generated language is concise, fluent, and semantically generic. As shown in Table 3(b), all models successfully produced relevant and coherent outputs, but GPT-4o-Vision received the highest scores for identifying and localizing temporal patterns. GPT-4o-Text produced fluent language but often misstated underlying trends, while ChatTS generated structurally accurate but sometimes overly literal or low-abstraction descriptions. Inter-annotator agreement was high, with Cohen's Kappa and Krippendorff's Alpha exceeding 0.8 for all criteria. As a complementary evaluation, we additionally report GPT-5-based entailment metrics for generated descriptions in Table 19. Across all three evaluation methods—DeBERTa NLI, GPT-5 judging, and human

annotation—the same relative model ranking is preserved, demonstrating that our generation evaluation is robust to evaluator choice.

### 4.2.3. HOW ROBUST ARE FOUNDATION MODELS AGAINST REALISTIC PERTURBATIONS?

We next evaluate how robust model performance on `BEDTime` is to realistic perturbations, an essential consideration for any system claiming to "describe" or "understand" time series. In real-world deployments, sequences may vary in length, or exhibit missing values, that challenge descriptive precision. To this end, we conduct **six** robustness tests designed to probe a models' generalization capabilities beyond the clean, fully observed inputs used in our main benchmark.

First, we assess robustness to increased sequence length by linearly interpolating time series to increase resolution while preserving core properties. This allows us to evaluate whether model descriptions remain consistent as inputs become longer. Second, we introduce structured missingness via uniform random masking, simulating data sparsity that is common in practical settings. We additionally evaluate the effects of amplitude scaling on model performance at five levels, where each level corresponds to a scalar multiplier applied to every element of the time series, and additive Gaussian noise by applying nine levels of perturbation (measured by standard deviation). For VLMs, we further introduce a plot-resolution perturbation by systematically degrading the visual fidelity of rendered time series plots via controlled pixelation, while keeping the underlying time series fixed, in order to isolate their sensitivity to image quality. Lastly, we also test the effect of chain-of-thought prompting strategy (Wei et al., 2023) on LLM performance.

**Effect of Varying the Length of the Time Series.** We observe that recognition and differentiation accuracy for text-only LLMs declines as sequence length increases (see Figure 2A and 2B), a trend consistent with known limitations of Transformer architectures. This degradation stems from three key factors: (1) the $\mathcal{O}(L^2)$ cost of self-attention with respect to input length $L$, which limits usable context via truncation or approximation (Vaswani et al., 2023); (2) attention diffusion, wherein softmax weights flatten over longer inputs, reducing focus on salient time steps; and (3) poor generalization of positional encodings beyond pretraining lengths, which distorts temporal localization (Gruver et al., 2023; Fons et al., 2024; Merrill et al., 2024). These limitations are most pronounced in smaller open-source models like Llama, which exhibit sharp drops in performance with input length, whereas larger models such as GPT and Gemini variants decline more gradually. Prior work further shows that such degradation is amplified by position bias, especially when target values appear late in

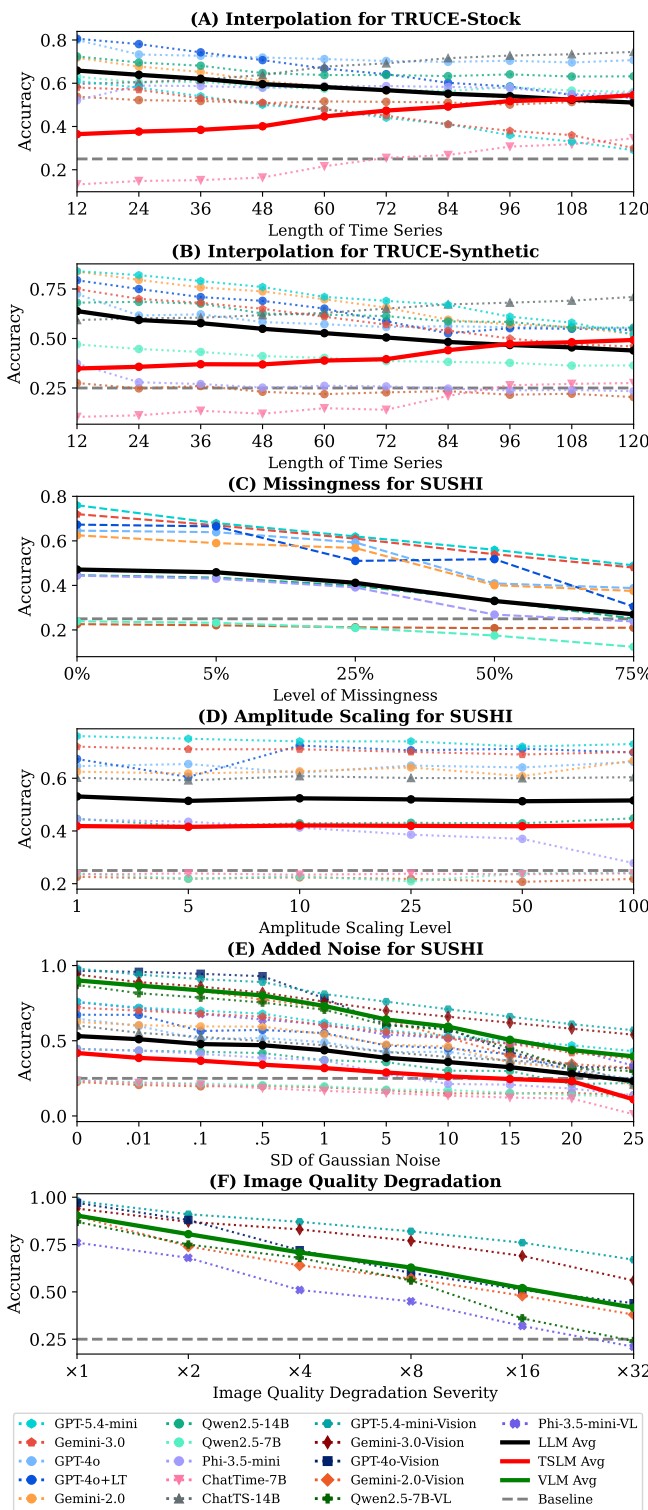

*Figure 2.* Robustness of LMs to **five** realistic perturbations for Recognition and Differentiation. Accuracy decline for LLMs as time series grow longer or contain more missing values. As signals get noisier performance decline. LMs are relatively robust to amplitude scaling. For VLMs, accuracy decreases as plot resolution degrades. DTW distance is used to pick distractors. For Complete Results refer to Figure 6

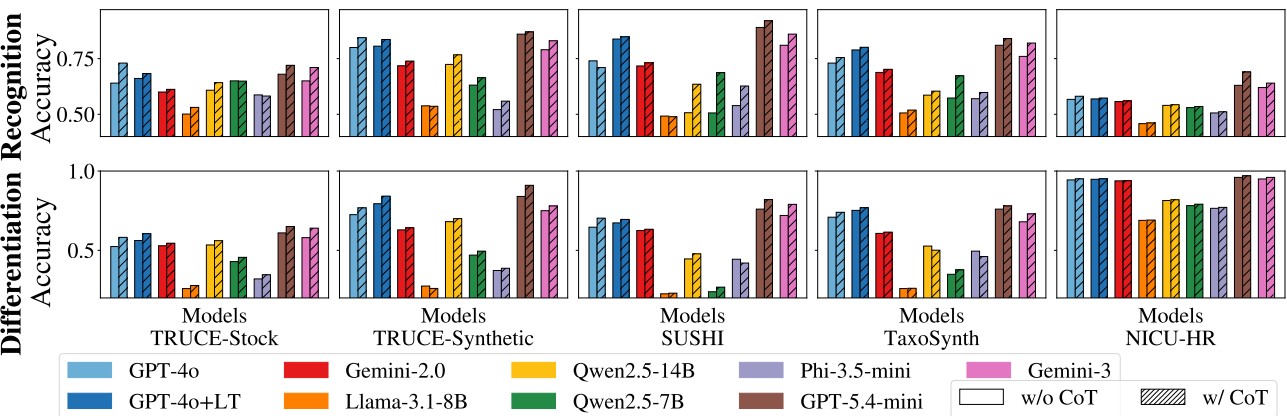

*Figure 3.* Impact of CoT prompting on LLMs' accuracy across five datasets and two tasks. CoT consistently improves both recognition and differentiation performance, with the largest gains observed on the differentiation task—especially for proprietary models. DTW distance is used to pick dissimilar descriptions.

the sequence, and by the inability of standard tokenizations to preserve long-range temporal structure. In contrast, both ChatTS and ChatTime improve with increasing sequence length, suggesting that TSLMs benefit from interpolation and are more robust to longer time series inputs. These hold for both Recognition and Differentiation (see Fig. 6).

Interpolating time series increases floating-point precision, which drastically inflates token counts for LLMs. This raises a natural question: is the observed performance drop primarily due to tokenization overhead rather than sequence length itself? To test this, we scaled interpolated values by 100 to convert floats into integers and evaluated whether this improved model accuracy. From Figure 7 we see that scaling does aid model performance across all models except Qwen2.5-14B, whose performance declines. We attribute this to Qwen2.5-14B's attention diffusion and weaker reliance on discrete token structure: its larger key-value head count and higher parameter capacity encourage more distributed representations, though the model shows limited gains on number-sensitive tasks (Yang et al., 2025).

**Effect of Missing Data.** We next measure the effect of missing values on model performance. To simulate real-world data sparsity, we introduce missingness by uniformly at random masking out a fraction of each time series, replacing those entries with NaN, at four levels (5 %, 25 %, 50 %, and 75 %). We then evaluate our same suite of LLMs under each missingness condition. The Differentiation results are shown in Figure 2C. Missing values degrade performance across all models and datasets. Accuracy remains relatively stable up to 25% data missingness, particularly for LLMs, but drops sharply once 50% or more of the time series is masked. This trend holds for both Recognition and Differentiation tasks (see Fig. 6).

**Effect of Amplitude Scaling.** We evaluate the effect of amplitude scaling on model performance by multiplying

each time series by a scalar at five levels (5, 10, 25, 50, 100). We then evaluate our LLM suite under each scaling condition. The Differentiation results are shown in Figure 2D. Both LLMs and TSLMs are broadly robust to amplitude scaling, with only minor changes in accuracy across levels. Notably, LLMs show slight improvements at the highest scaling factor, likely due to enhanced visibility of features in scaled signals. This trend holds across both recognition and differentiation tasks (see Fig. 6).

**Effect of Gaussian Noise.** We measure the effect of additive Gaussian noise on model performance by perturbing each time series with noise at nine levels (0, 0.01, 0.1, 0.5, 1, 5, 10, 15, 20, 25), defined by the standard deviation of the noise distribution. We then evaluate our models suite containing LLMs, VLMs, and TSLMs under each noise condition. The Differentiation results are shown in Figure 2E. Performance consistently degrades as noise increases. Text-only models' performance rapidly falls below task-specific random baselines, whereas multimodal models (VLMs and TSLMs) show greater robustness, with slower degradation under higher levels of additive Gaussian noise. These trends hold for both Recognition and Differentiation (see Fig. 6).

**Effect of Image Quality.** To evaluate the impact of plot resolution on VLM performance, we conduct a controlled pixelation study that isolates visual fidelity as the sole varying factor. We systematically degrade plot resolution while keeping the underlying time series fixed, following a standard downscale-upscale protocol. Starting from the original plot, we downsample each image by factors $f \in \{2, 4, 8, 16, 32\}$ and then upsample back using nearest-neighbor interpolation, yielding six fidelity levels. Figure 2F reports accuracy across all 6 pixelation levels for each VLM on Differentiation. VLM accuracy degrades monotonically as pixelation severity increases. GPT vision variants are the most robust, maintaining strong performance through moderate degradation ($\times 8-\times 16$). Gemini-2.0-Vision follows

a similar but weaker trend, degrading more rapidly under heavier corruption. In contrast, Qwen2.5-VL-7B and Phi-3.5-Vision are substantially less robust: Qwen2.5-VL-7B drops sharply at moderate pixelation ($\times 4$), while Phi-3.5-Vision deteriorates faster at extreme levels. These hold for both Recognition and Differentiation (see Fig. 6).

**Effect of Chain-of-Thought.** We evaluate the impact of Chain-of-Thought (CoT) prompting on model performance by testing all LLMs from our model suite on both Recognition and Differentiation tasks, with and without CoT prompting across all five datasets. The results are shown in Figure 3. Across every model and dataset, CoT prompting yields a consistent gain in performance for LLMs, though the overall effect is modest, with a mean absolute accuracy increase of +0.03 across all model–dataset pairs for both tasks. Focusing on the two models showing gains exceeding 5%, we identify two distinct sources of improvement. First, the largest relative gains come from reduced refusal rates: Qwen2.5-7B improves by +11.85% (R) and +6.58% (D) on average, with the highest gains on SUSHI, where refusal rates drop from 24.7% to 11.8% (R) and 27.2% to 19.4% (D). Second, GPT-4o shows notable gains on TRUCE-Stock, where CoT reduces systematic positional bias toward the first available option by 18.1% (R) and 11.7% (D). More broadly, larger proprietary models benefit more from CoT than smaller open-weight models, closing much of the gap to VLMs. These findings demonstrate that CoT is an effective strategy for improving LLM performance in time series description tasks, with the largest gains attributable to reduced refusal rates and mitigated positional biases.

## 5. Conclusions

We introduce BEDTime, a unified benchmark for evaluating language models on structural description of time series, a fundamental capability for temporal reasoning. BEDTime defines three tasks and unifies five public datasets, enabling head-to-head evaluation of LLMs, VLMs, and TSLMs with over 40,000 time series–description pairs, more than 90% of which are real-world signals. Our experiments show that models across modalities, families, and sizes still have clear room for improvement on every task, especially under real-world perturbations. At the same time, VLMs demonstrate notable successes, underscoring the value of structural representations for time series understanding. Overall, BEDTime provides a unified and extensible framework for evaluating time series description; we release its code and data to ensure reproducibility and support evaluation of future models and emerging time series–text datasets.

## 6. Limitations and Future Directions

Multi-modal time series–text understanding and reasoning is an emerging research direction, and systematic frameworks for evaluating core model capabilities remain underdeveloped. This is partially due to the lack of standardized evaluation protocols and high-quality real-world datasets, so naturally BEDTime has limitations that can inspire future work. First, while over 90% of our benchmark data points are real-world signals, we also rely on synthetic data, as they are the focus of multiple related prior works. Second, BEDTime studies univariate time series, which are widely studied in the time series literature and underpin many real-world tasks across domains such as healthcare, finance and physical systems. Future work may extend time series description benchmarks to multivariate settings. Third, BEDTime relies on general-purpose NLI models for automated assessment, which can be sensitive to numerical inputs. Although we incorporate GPT-5 to mitigate this issue, automated evaluation may still introduce residual variability. Fourth, our empirical findings are shaped by the models and model families evaluated. In particular, only two publicly-available TSLMs currently support our evaluation setting.

## Acknowledgements

TH was supported by a CapitalOne Faculty Fellowship and the University of Virginia's National Security Data & Policy Institute through the U.S. Department of Defense Contracting Activity #2024-24070100001. The compute resources for this work were supported by Microsoft's Accelerating Foundation Model Research program. We thank the University of Virginia's High Performance Computing team for providing excellent computing resources. We also thank Mike Merrill, Matt Landers, and Marco Gutierrez for their valuable feedback on earlier drafts of this paper.

## Impact Statement

This paper introduces BEDTime, a benchmark for evaluating language models on time-series description. Our goal is to advance the field of machine learning by enabling more systematic evaluation and comparison of methods on time-series–language tasks. There are many potential societal consequences of our work, none of which we feel must be specifically highlighted here.

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

# Appendix

## A. Extended Literature Review

To ground our discussion of multi-modal time series reasoning, we first situate BEDTime within the broader landscape of existing benchmarks. Table 4 compares representative synthetic, empirical, QA-style, and multimodal benchmarks in terms of their tasks, data modalities, evaluation protocols, and limitations, highlighting how prior benchmarks emphasize question answering, forecasting, or domain-specific reasoning rather than foundational time series description skills.

*Table 4.* Comparison of major time-series reasoning, QA, captioning, and multimodal benchmarks with BEDTime.

| Benchmark / Dataset | Tasks Evaluated | Data Type | Metrics | Key Contributions | Limitations | BEDTime's Novelty |
|---|---|---|---|---|---|---|
| **Merrill et al., 2024** (Merrill et al., 2024) | Etiological reasoning; TS question answering; context-aided forecasting with multiple-choice answers | Fully synthetic univariate TS with associated text scenarios; ~8.7k TS–text pairs and ~230k MCQs | MCQ accuracy for reasoning; forecast error metrics for numeric predictions | Defines three TS reasoning task families and a large synthetic MCQ suite; shows LLMs struggle at zero-shot TS reasoning, with performance often near random | All TS are synthetic with no real-world signals; TS–text relationships not grounded in observational noise or human-authored descriptions; no real-world robustness tests | BEDTime uses real TRUCE-Stock with human crowd descriptions plus three synthetic/template datasets; focuses on visually grounded descriptions; includes human evaluation and robustness analysis revealing fundamental description failures |
| **TimeSeries Exam** (Cai et al., 2024) | Pattern recognition; noise understanding; anomaly detection; similarity analysis; causality analysis via MCQs | Fully synthetic univariate TS from controlled base patterns, compositions, and transformations; grouped by TS concept | MCQ accuracy per category for pattern, anomaly, noise, similarity, causality | Provides a carefully controlled synthetic domain-agnostic exam for TS concepts; fine-grained diagnostic breakdown | Only synthetic TS; MCQ-only evaluation; no free-form language generation; no TS–text description pairs; no real data | BEDTime keeps domain-agnostic structure but includes real + synthetic TS–description pairs; adds recognition, MCQ differentiation, and open generation; evaluates numeric TS, plots, and text with NLI + human scoring rather than MCQ accuracy only |
| **Time-MQA** (Kong et al., 2025) | Unified TS QA over forecasting, imputation, anomaly detection, classification, and reasoning QA (MCQ, T/F, open QA) | TSQA dataset with ~192k QA pairs across 12 domains (healthcare, finance, energy, traffic, IoT, etc.); real + synthetic TS with associated text (benchmarks, transcripts) | Regression errors for forecasting/imputation; accuracy/F1 for anomaly detection, classification, QA | Proposes TSQA dataset and Time-MQA formulation; shows continual pretraining improves TS reasoning | Many QA items and explanations are GPT-4o-generated; open-answer evaluation relies on generic text metrics; uses numeric TS + text only (no plot modality) | BEDTime centers on TS–description pairs from human/crowd/template/synthetic sources (not dominated by GPT-4); adds explicit plot modality; compares LLM vs VLM vs TSLM; focuses on foundational TS description skills that underlie QA |
| **ITFormer** (Wang et al., 2025b) | TS understanding, perception, reasoning, and maintenance decision QA for aero engines | QA pairs from NCMAPSS aero-engine multivariate TS (32 channels, 10-min segments); >110k QA pairs | BLEU, ROUGE-L for open QA; accuracy/F1 for classification-type QA | First large-scale real-world TS-QA dataset for aero-engine monitoring; defines a four-level TS understanding taxonomy; proposes ITFormer encoder + LLM | Restricted to one industry/domain; questions are programmatically generated and task-specific rather than generic TS pattern descriptions | BEDTime is domain-agnostic across five datasets; evaluates simple, broad TS description tasks; compares many foundation models across modalities; includes systematic robustness tests |

| MTBench (Chen et al., 2025) | Multimodal TS–text reasoning: forecasting; semantic trend classification; technical indicator prediction; news-driven QA; predicting correlation between news and future trends | Paired TS + text for finance and weather: stock series with news and financial reports; weather TS with textual weather reports from 50 stations | Regression metrics for numeric tasks; accuracy and related scores for QA, trend, and correlation prediction | First benchmark on TS–text causal and temporal reasoning; aligns reports/news with TS to study cross-modal relationships | Domains limited to finance + weather; correlation labels partly generated via GPT-4; causal reasoning reduced to classification labels | BEDTime is domain-agnostic; focuses on generic TS patterns; uses pre-existing human/template descriptions rather than new GPT labels; emphasizes low-level TS description abilities transferable across domains |
| --- | --- | --- | --- | --- | --- | --- |
| TADACap (Fons et al., 2025) | TS captioning on plots; retrieval-based captioning vs VLM captioning | Line plots from four datasets (Real-COVID, RealKnee, SynthPhysics, SynthStocks); domain-specific captioned plot images | ROUGE-L; CIDEr; SPICE; SPIDEr | Introduces domain-aware TS captioning; provides four plot-caption datasets; compares retrieval vs VLM captions | Focused on specific domains; only evaluates plot images (no numeric TS); lacks unified benchmark spanning modalities/datasets | BEDTime unifies five datasets into one benchmark with numeric series, plots, and text; defines three generic TS description tasks; compares LLMs, VLMs, and TSLMs directly |
| CaTS-Bench (Zhou et al., 2025) | TS captioning; MCQ TS–caption matching; plot–caption matching; comparative reasoning QA | ~11 real-world datasets containing numeric TS, metadata, line plots, and LLM-generated captions; associated MCQ items | F1 and similarity (DeBERTa); BLEU; ROUGE-L; METEOR; numeric fidelity metrics (mean/var/min/max) | First large-scale context-aware TSC benchmark; introduces numeric fidelity checks; adds diagnostic QA tasks | Captions mostly from one oracle LLM pipeline (stylistically homogeneous); visuals restricted to standard line plots; VLMs underuse plots | BEDTime aggregates five heterogeneous description sources (crowd, template, synthetic); evaluates recognition, differentiation, and generation via NLI + human scores; includes robustness tests and multi-modality comparisons |
| QuAnTS (Divo et al., 2025) | Time-series question answering (TSQA): binary (T/F), multiple-choice, and open-ended QA conditioned on explicit natural-language questions | Fully synthetic multivariate human-motion time series generated via motion diffusion; skeleton trajectories paired with templated and paraphrased QA pairs | Accuracy/F1 for binary and MCQ; ROUGE, METEOR, and LLM-as-judge scores for open-ended QA; includes human reference performance | Introduces the first large-scale TSQA benchmark; defines a comprehensive taxonomy of 46 question types; enables controlled evaluation of temporal reasoning when questions specify what to attend to | QA-centric framing provides strong linguistic scaffolding; restricted to a single synthetic domain (human motion); no plot modality; does not evaluate unprompted or holistic time-series description ability | BEDTime removes question scaffolding to evaluate foundational TS description skills; uses heterogeneous real + synthetic datasets with human and crowd descriptions; adds explicit plot modality; evaluates recognition, differentiation, and generation with robustness and cross-modality comparisons |
| BEDTime (Ours) | Recognition (T/F), differentiation (MCQ), open-ended TS description generation; robustness tests (length, missingness, amplitude scaling, noise, CoT prompting) | 46,843 univariate TS–description pairs (with more than 90% real-world signals) from TRUCE-Stock, TRUCE-Synthetic, TaxoSynth, SUSHI and NICU-HR; human, crowd, template, synthetic; supports numeric TS, plots, and text across LLM/VLM/TSLM | Accuracy for recognition & differentiation; NLI entailment (bidirectional) for generation; 6-criterion human evaluation; robustness metrics | First unified benchmark for TS description across three tasks and three modalities; adapts five datasets into one; head-to-head comparisons of LLMs, VLMs, TSLMs; extensive robustness study; releases full code+data | Limited to univariate TS; much of the data is synthetic except TRUCE-Stock; generation evaluation uses general-purpose NLI models not TS-specific | Defines the first unified framework and evaluation protocol for TS description tasks, enabling all prior datasets to be mapped onto a consistent structure |

While Table 4 contrasts benchmarks and datasets, we next examine model-level applicability. In particular, many recent time series–language models differ substantially in how linguistic inputs and outputs is incorporated ranging from auxiliary prompts and retrieval supervision to decision-conditioned explanations which affects whether they can be meaningfully evaluated on BEDTime 's description recognition, differentiation, and generation tasks.

Table 5 summarizes the time series–language models we considered when constructing our evaluation suite and clarifies the architectural and practical reasons many are not directly compatible with BEDTime. This analysis explains why, despite the breadth of existing TSLMs, only a small subset can be evaluated on BEDTime 's description-focused tasks.

*Table 5.* Summary of time series–language models considered in our comparison and their applicability to BEDTime. The table documents the models examined when constructing our evaluation suite and the architectural reasons many are not directly compatible with BEDTime's description recognition, differentiation, and generation tasks. Based on these criteria and the availability of reproducible implementations, our empirical evaluation focuses on the two TSLMs (ChatTS and ChatTime) that can be meaningfully evaluated on BEDTime. We also reviewed two additional recent TSLMs designed explicitly for joint time series and text question answering (Chow et al., 2024; Trabelsi et al., 2025); however, neither releases code, pretrained weights, or the proprietary or synthetic datasets required for reproduction, making evaluation on BEDTime infeasible.

| Model | Takes TS + Text? | Main Abilities | Natural Language Generation? | Why Not Usable in BEDTime |
|---|---|---|---|---|
| **TEST** (Sun et al., 2024) | Yes | Classification, forecasting | No | Never trained to interpret natural language descriptions of time series or to produce text outputs. The architecture routes time series through an LLM solely to improve numeric prediction, making it unsuitable for description recognition, differentiation, or generation. |
| **Time-LLM** (Jin et al., 2024) | Yes | Forecasting | No | The LLM backbone is reprogrammed with projection heads for numeric forecasting only. There is no mechanism for semantic TS–text matching or language decoding, preventing description recognition, differentiation, or generation. |
| **CALF** (formerly LLaTa) (Liu et al., 2025a) | Yes | Forecasting | No | All outputs are numeric. The model lacks any language generation or description understanding component and therefore cannot generate or identify time series descriptions. |
| **UniTime** (Liu et al., 2024) | Yes | Forecasting | No | Language tokens act as domain or task identifiers rather than semantic descriptions. The model predicts future numeric values only and does not support description generation or comparison. |
| **UNITS** (Gao et al., 2024) | No | Forecasting, classification, imputation, anomaly detection | No | A time-series-only model with no text interface. It cannot ingest text, compare time series against descriptions, or generate language, and thus fails all BEDTime tasks. |
| **CLaSP** (Ito et al., 2025) | Yes | Contrastive alignment, retrieval | Retrieval only | Learns joint embeddings for nearest-neighbor caption retrieval rather than text generation. It cannot evaluate description correctness or generate new descriptions, limiting it to retrieval-based alignment only. |
| **TimeVLM** (Zhong et al., 2025) | Yes | Multimodal forecasting | No | Although it constructs textual prompts (statistics, domain context) to aid forecasting, these texts are auxiliary inputs, not semantic targets. The model never evaluates, compares, or generates free-form descriptions of time series structural properties, and outputs are strictly numeric forecasts; therefore it cannot perform BEDTime 's recognition, differentiation, or open-ended description generation tasks. |
| **TimeXL** (Jiang et al., 2025) | Yes | Multimodal prediction with explanations | Yes (rationales only) | It generates decision conditioned rationales that explain why a specific downstream prediction (e.g., classification or regression) was made within a closed-loop, agentic system. These texts are intrinsically tied to task labels and prediction objectives, rather than to the standalone structural properties of the time series itself. As a result, TimeXL cannot perform BEDTime's core tasks of (i) recognizing whether an arbitrary description matches a time series, (ii) differentiating correct from incorrect descriptions, or (iii) generating label-agnostic, generic natural-language descriptions of structural time-series characteristics. |
| **SensorLM** (Zhang et al., 2025) | Yes (sensor TS + text) | Sensor understanding, activity recognition, captioning | Yes | Captions are generated under strong wearable-specific and activity-semantic assumptions (e.g., sleep, workouts), rather than generic structural descriptions of abstract time series. SensorLM is trained to interpret embodied human sensor data, not to recognize, differentiate, or generate domain-agnostic descriptions of time series signals required by BEDTime. |
| **Timer-XL** (Liu et al., 2025b) | No | Unified long-context forecasting | No | It is a decoder-only time-series Transformer trained via multivariate next-token prediction over numeric patch tokens. It has no language interface, no text encoder or decoder, and cannot ingest, compare, or generate natural-language descriptions, making it incompatible with all BEDTime tasks. |

Together, these benchmark and model level comparisons motivate our experimental focus on models that can directly map between time series and natural language descriptions across modalities.

## B. Dataset Description

To support our benchmark tasks, we unify and reformat five public datasets containing paired time series and natural language descriptions. These datasets vary in realism, sequence length, and linguistic complexity, and are summarized in Table 1. These datasets described in Section 3.2 support the recognition, differentiation, and generation tasks, though each dataset is used selectively depending on task requirements and annotation structures.

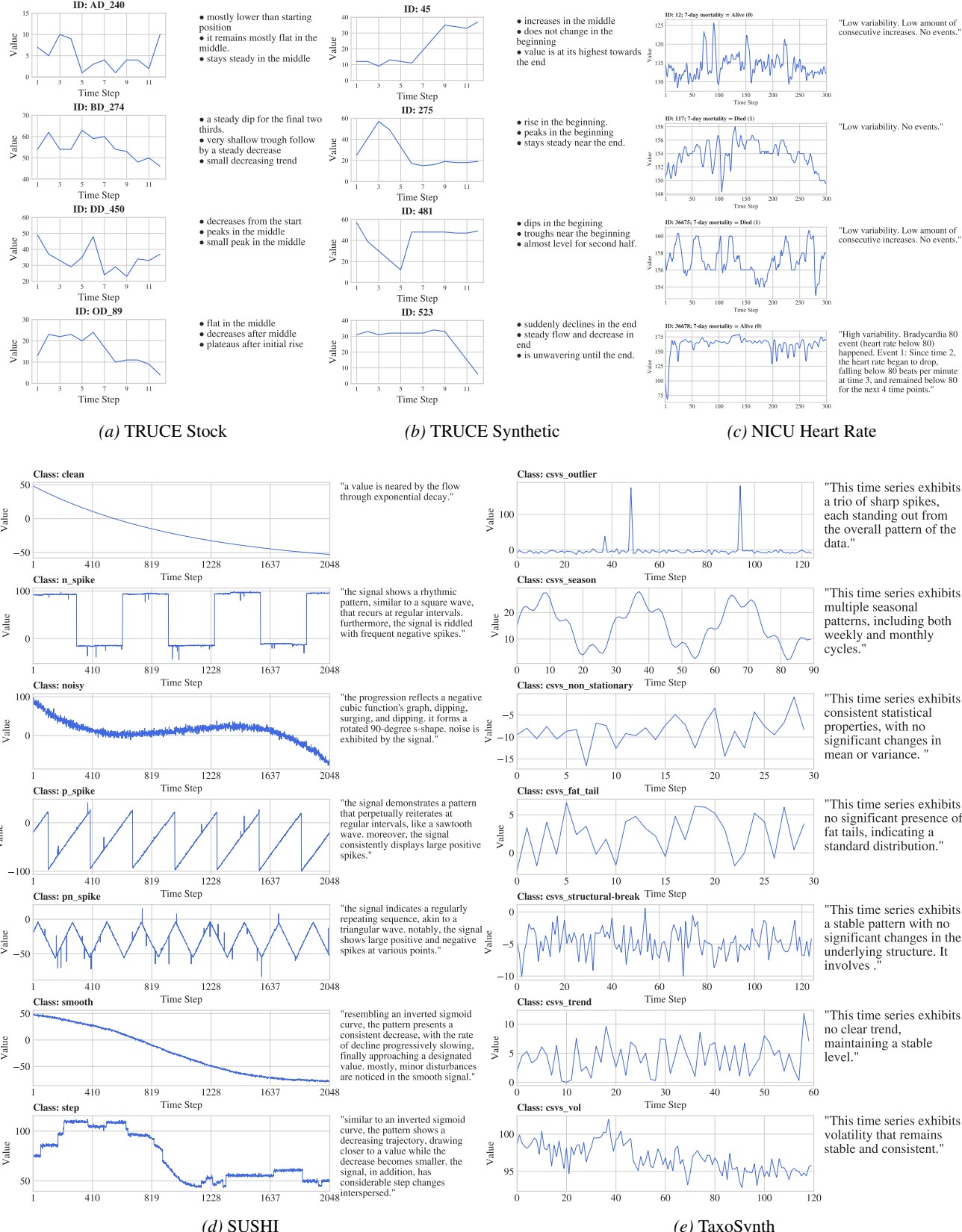

*Figure 4.* Representative time series and descriptions from each dataset used in BEDTime. These examples illustrate variations in sequence length, description style, and data generation method.

Figure 4 illustrates representative samples from each dataset, showcasing the diversity in time series patterns and caption styles—from concise, crowd-sourced descriptions of financial data to longer, templated descriptions of synthetically generated sequences. Together, the table and examples highlight the breadth of data modalities and linguistic forms evaluated by BEDTime.

## C. Sampling Incorrect Options

To support both True/False and Multiple Choice formats in the Recognition and Differentiation tasks, we construct contrastive examples by selecting negative descriptions using four distinct strategies:

- **Caption-based similarity (Sentence-BERT):** We compute cosine similarity over Sentence-BERT embeddings and select descriptions that are semantically dissimilar to the reference.

- **Dynamic Time Warping (DTW):** We measure alignment costs between time series and choose those with the highest DTW distance from the input.

- **Euclidean Distance:** We identify point-wise dissimilar series based on maximum L2 distance.

- **Longest Common Subsequence (LCSS):** We retrieve sequences with minimal overlapping subsequences, prioritizing structural dissimilarity.

Only the Sentence-BERT strategy operates over natural language annotations; the other three assess distance directly in time series space. When multiple annotations exist for a given time series, we randomly sample one for evaluation. Negative samples are selected to be maximally dissimilar, simplifying the contrastive setup and providing an upper-bound estimate of model performance. This design ensures that the benchmark evaluates models' ability to reject clearly incorrect options before advancing to more fine-grained reasoning.

## D. Prompts

We include the exact prompts used for each BEDTime task below. These prompts were used across models for consistent evaluation and follow the formats outlined in Section 3.1.

**Task 1: Recognition.**

> You are tasked with verifying if the provided description accurately describes the given time series.
> Please follow these instructions carefully:
> 1. Review the description: {description}.
> 2. Analyze the time series: {series}.
> 3. Determine if the annotation precisely matches the pattern depicted in the time series.
> Respond with **True** if the given description accurately describes the time series.
> Respond with **False** if it does not. Avoid providing any additional comments or explanations.

**Task 2: Differentiation.**

> Carefully analyze the given time series and choose the single best option that most accurately describes its pattern.
> Follow these rules strictly:
> 1. Read all options before deciding.
> 2. Only output the chosen option, highlighted as A, B, C, or D.
> 3. Avoid adding extra text or explanations.
> **Time series:** {series}
> **Options:**
> - A: {option_1}
> - B: {option_2}

- C: {option_3}
- D: {option_4}

**Task 3: Open Generation.**

You are tasked with generating a textual description of the structural properties of the provided time series.
Please follow these instructions carefully:
1. Analyze the given time series data: {series}.
2. Identify and describe the most prominent visual features or patterns observed in the time series.
Consider characteristics such as trends, seasonality, anomalies, or significant changes.
Your response should be a concise textual description of the most pronounced structural properties of the time series.
Avoid including unnecessary details or unrelated commentary.

**Note:** The format of the time series input varies based on model modality. LLMs receive a comma-separated string of numeric values. VLMs receive a `matplotlib`-rendered image of the time series or a base64-encoded image string of the same. TSLMs are provided with the raw sequence as a `NumPy` array or Python list of floats.

## E. Recognition and Differentiation: Full Results Across All Sampling Strategies

To support the findings presented in Section 4.2.1, we report full accuracy and F1 score (F1 weighted in the case of differentiation) across all datasets and all four negative sampling strategies (Sentence-BERT, DTW, Euclidean, and LCSS) for the *Recognition* (True/False) and *Differentiation* (Multiple Choice) tasks. Tables 6, 7, 8, 9, 10, 11, 12, 13, 14, and 15 provide per-dataset, per-sampling performance scores.

*Table 6.* Performance of Various Language and Vision-Language Models on TRUCE-Stock Dataset in the Recognition Setting using 4 Different Negative Sampling Techniques

| Model | S-Bert Embeddings with Cosine Similarity | | DTW Distance | | Euclidean Distance | | Longest Common Subsequence | |
|---|---|---|---|---|---|---|---|---|
| | Accuracy | F1 Score | Accuracy | F1 Score | Accuracy | F1 Score | Accuracy | F1 Score |
| **Language Models** | | | | | | | | |
| GPT-5.4-mini | **0.712** | **0.713** | **0.679** | **0.671** | **0.699** | **0.699** | **0.688** | **0.686** |
| Gemini-3-flash | 0.709 | 0.705 | 0.653 | 0.641 | 0.648 | 0.643 | 0.677 | 0.673 |
| GPT-4o | 0.610 | 0.581 | 0.640 | 0.556 | 0.592 | 0.678 | 0.592 | 0.527 |
| GPT-4o (LLMTime) | 0.703 | 0.700 | 0.661 | 0.661 | 0.692 | 0.679 | 0.616 | 0.614 |
| Gemini-2.0-Flash | 0.607 | 0.546 | 0.600 | 0.528 | 0.585 | 0.557 | 0.534 | 0.472 |
| Llama-3.1-8B-Instruct | 0.501 | 0.334 | 0.501 | 0.334 | 0.501 | 0.334 | 0.501 | 0.334 |
| Qwen2.5-14B-Instruct-1M | 0.642 | 0.615 | 0.608 | 0.587 | 0.606 | 0.585 | 0.522 | 0.512 |
| Qwen2.5-7B-Instruct-1M | 0.674 | 0.656 | 0.650 | 0.635 | 0.643 | 0.630 | 0.519 | 0.516 |
| Phi-3.5-mini-instruct | 0.526 | 0.389 | 0.587 | 0.519 | 0.520 | 0.386 | 0.519 | 0.385 |
| **Vision-Language Models** | | | | | | | | |
| GPT-5.4-mini (with image inputs) | **0.853** | **0.850** | **0.791** | **0.790** | **0.766** | **0.764** | **0.878** | **0.872** |
| Gemini-3-flash (with image inputs) | 0.824 | 0.818 | 0.730 | 0.727 | 0.754 | 0.748 | 0.818 | 0.814 |
| GPT-4o (with image inputs) | 0.842 | 0.840 | 0.750 | 0.719 | 0.760 | 0.727 | 0.840 | 0.820 |
| Gemini-2.0-Flash (with image inputs) | 0.793 | 0.786 | 0.682 | 0.648 | 0.683 | 0.648 | 0.675 | 0.648 |
| Qwen2.5-VL-7B-Instruct | 0.713 | 0.695 | 0.609 | 0.539 | 0.609 | 0.539 | 0.573 | 0.512 |
| Phi-3.5-vision-instruct | 0.824 | 0.822 | 0.686 | 0.685 | 0.679 | 0.679 | 0.612 | 0.611 |
| **Language Models for Time Series** | | | | | | | | |
| ChatTime-7B | 0.050 | 0.049 | 0.102 | 0.099 | 0.099 | 0.094 | 0.093 | 0.091 |
| ChatTS-14B | **0.414** | **0.393** | **0.423** | **0.404** | **0.405** | **0.398** | **0.453** | **0.438** |

Table 6 reports model performance on the TRUCE-Stock dataset in the Recognition setting, using all four contrastive sampling strategies. As the only real-world dataset in the benchmark, TRUCE-Stock presents the greatest challenge, with lower overall performance across models. GPT-5.4-mini performs best among language-only models, and GPT-5.4-mini with image input leads across all modalities, though with a smaller margin than in synthetic datasets. Most open-source

LLMs underperform across distractor types, with particularly poor results from smaller models such as Phi and Llama. Time-series-specific models show limited effectiveness here as well, with ChatTS underperforming relative to its base LLM counterpart, suggesting that its strengths may not translate as well to short, real-world financial sequences.

*Table 7.* Performance of Various Language and Vision-Language Models on TRUCE-Synthetic Dataset in the Recognition Setting using 4 Different Negative Sampling Techniques

| Model | S-Bert Embeddings with Cosine Similarity | | DTW Distance | | Euclidean Distance | | Longest Common Subsequence | |
|---|---|---|---|---|---|---|---|---|
| | Accuracy | F1 Score | Accuracy | F1 Score | Accuracy | F1 Score | Accuracy | F1 Score |
| **Language Models** | | | | | | | | |
| GPT-5.4-mini | **0.822** | **0.820** | **0.865** | **0.863** | **0.719** | **0.718** | **0.822** | **0.818** |
| Gemini-3-flash | 0. 789 | 0.784 | 0.794 | 0.791 | 0.709 | 0.708 | 0.798 | 0.794 |
| GPT-4o | 0.790 | 0.760 | 0.800 | 0.792 | 0.691 | 0.654 | 0.780 | 0.776 |
| GPT-4o (LLMTime) | 0.806 | 0.804 | 0.806 | 0.802 | 0.696 | 0.695 | 0.802 | 0.802 |
| Gemini-2.0-Flash | 0.724 | 0.699 | 0.718 | 0.711 | 0.675 | 0.669 | 0.707 | 0.703 |
| Llama-3.1-8B-Instruct | 0.535 | 0.409 | 0.538 | 0.412 | 0.537 | 0.411 | 0.522 | 0.400 |
| Qwen2.5-14B-Instruct-1M | 0.819 | 0.817 | 0.724 | 0.724 | 0.710 | 0.710 | 0.715 | 0.715 |
| Qwen2.5-7B-Instruct-1M | 0.758 | 0.756 | 0.631 | 0.631 | 0.623 | 0.622 | 0.659 | 0.659 |
| Phi-3.5-mini-instruct | 0.588 | 0.519 | 0.521 | 0.386 | 0.581 | 0.514 | 0.538 | 0.482 |
| **Vision-Language Models** | | | | | | | | |
| GPT-5.4-mini (with image inputs) | **0.910** | **0.897** | **0.884** | **0.878** | **0.897** | **0.882** | **0.935** | **0.912** |
| Gemini-3-flash (with image inputs) | 0.843 | 0.841 | 0.845 | 0.841 | 0.786 | 0.784 | 0.894 | 0.889 |
| GPT-4o (with image inputs) | 0.893 | 0.890 | 0.830 | 0.838 | 0.880 | 0.800 | 0.920 | 0.868 |
| Gemini-2.0-Flash (with image inputs) | 0.741 | 0.735 | 0.724 | 0.716 | 0.752 | 0.751 | 0.723 | 0.712 |
| Qwen2.5-VL-7B-Instruct | 0.607 | 0.537 | 0.716 | 0.699 | 0.717 | 0.699 | 0.660 | 0.648 |
| Phi-3.5-vision-instruct | 0.722 | 0.720 | 0.658 | 0.634 | 0.660 | 0.637 | 0.673 | 0.653 |
| **Language Models for Time Series** | | | | | | | | |
| ChatTime-7B | 0.174 | 0.172 | 0.132 | 0.130 | 0.132 | 0.130 | 0.162 | 0.160 |
| ChatTS-14B | **0.509** | **0.491** | **0.597** | **0.591** | **0.518** | **0.512** | **0.460** | **0.457** |

Table 7 presents Recognition performance on the TRUCE-Synthetic dataset. As a structured synthetic benchmark with short sequences and well-defined up/down patterns, this dataset yields higher overall model performance than TRUCE-Stock. GPT-5.4-mini again leads among LLMs, while GPT-5.4-mini with image input achieves the highest scores across all distractor types, including near-perfect F1. ChatTS performs notably better here than on TRUCE-Stock, highlighting its alignment with synthetic pattern recognition.

*Table 8.* Performance of Language and Vision-Language Models on SUSHI Dataset in the Recognition Setting using 4 Different Negative Sampling Techniques

| Model | S-Bert Embeddings with Cosine Similarity | | DTW Distance | | Euclidean Distance | | Longest Common Subsequence | |
|---|---|---|---|---|---|---|---|---|
| | Accuracy | F1 Score | Accuracy | F1 Score | Accuracy | F1 Score | Accuracy | F1 Score |
| **Language Models** | | | | | | | | |
| GPT-5.4-mini | **0.913** | **0.907** | **0.889** | **0.887** | **0.844** | **0.836** | **0.887** | **0.884** |
| Gemini-3-flash | 0.867 | 0.861 | 0.814 | 0.813 | 0.733 | 0.733 | 0.814 | 0.808 |
| GPT-4o | 0.874 | 0.838 | 0.740 | 0.700 | 0.764 | 0.701 | 0.862 | 0.860 |
| GPT-4o (LLMTime) | 0.894 | 0.876 | 0.838 | 0.773 | 0.813 | 0.722 | 0.885 | 0.882 |
| Gemini-2.0-Flash | 0.775 | 0.765 | 0.717 | 0.647 | 0.694 | 0.615 | 0.809 | 0.779 |
| Llama-3.1-8B-Instruct | 0.509 | 0.362 | 0.492 | 0.354 | 0.493 | 0.354 | 0.504 | 0.356 |
| Qwen2.5-14B-Instruct-1M | 0.510 | 0.357 | 0.507 | 0.357 | 0.506 | 0.354 | 0.510 | 0.358 |
| Qwen2.5-7B-Instruct-1M | 0.507 | 0.352 | 0.506 | 0.351 | 0.508 | 0.354 | 0.506 | 0.352 |
| Phi-3.5-mini-instruct | 0.556 | 0.484 | 0.539 | 0.472 | 0.567 | 0.492 | 0.582 | 0.503 |
| **Vision-Language Models** | | | | | | | | |
| GPT-5.4-mini (with image inputs) | **0.984** | **0.984** | **0.973** | **0.971** | **0.958** | **0.957** | **0.987** | **0.982** |
| Gemini-3-flash (with image inputs) | 0.901 | 0.886 | 0.907 | 0.903 | 0.789 | 0.774 | 0.832 | 0.827 |
| GPT-4o (with image inputs) | 0.980 | 0.979 | 0.960 | 0.927 | 0.942 | 0.940 | 0.980 | 0.980 |
| Gemini-2.0-Flash (with image inputs) | 0.877 | 0.863 | 0.856 | 0.819 | 0.752 | 0.727 | 0.794 | 0.772 |
| Qwen2.5-VL-7B-Instruct | 0.613 | 0.545 | 0.612 | 0.544 | 0.611 | 0.543 | 0.613 | 0.545 |
| Phi-3.5-vision-instruct | 0.662 | 0.658 | 0.794 | 0.794 | 0.807 | 0.807 | 0.790 | 0.790 |
| **Language Models for Time Series** | | | | | | | | |
| ChatTime-7B | 0.321 | 0.319 | 0.279 | 0.276 | 0.261 | 0.257 | 0.301 | 0.300 |
| ChatTS-14B | **0.794** | **0.793** | **0.781** | **0.781** | **0.697** | **0.693** | **0.746** | **0.742** |

Table 8 reports Recognition results on the SUSHI dataset, which features long, synthetic time series with well-structured seasonal and trend components. Performance is uniformly higher here than on other datasets, with GPT-5.4-mini (with image input) achieving near-ceiling F1 scores across all distractor types. ChatTS performs particularly well in this setting, even surpassing many open-source LLMs, while smaller models such as Llama and Phi continue to underperform—highlighting the importance of both modality and model capacity when reasoning over long sequences.

*Table 9.* Performance of Various Language and Vision-Language Models on TaxoSynth Dataset in the Recognition Setting using 3 Different Negative Sampling Techniques

| Model | S-Bert Embeddings with Cosine Similarity | | DTW Distance | | Longest Common Subsequence | |
|---|---|---|---|---|---|---|
| | Accuracy | F1 Score | Accuracy | F1 Score | Accuracy | F1 Score |
| **Language Models** | | | | | | |
| GPT-5.4-mini | **0.877** | **0.876** | **0.814** | **0.814** | **0.913** | **0.909** |
| Gemini-3-flash | 0.791 | 0.789 | 0.763 | 0.762 | 0.718 | 0.716 |
| GPT-4o | 0.787 | 0.754 | 0.730 | 0.687 | 0.703 | 0.684 |
| GPT-4o (LLMTime) | 0.861 | 0.845 | 0.789 | 0.755 | 0.887 | 0.881 |
| Gemini-2.0-Flash | 0.720 | 0.694 | 0.688 | 0.633 | 0.662 | 0.614 |
| Llama-3.1-8B-Instruct | 0.514 | 0.367 | 0.506 | 0.364 | 0.508 | 0.362 |
| Qwen2.5-14B-Instruct-1M | 0.620 | 0.536 | 0.586 | 0.506 | 0.582 | 0.486 |
| Qwen2.5-7B-Instruct-1M | 0.612 | 0.529 | 0.573 | 0.492 | 0.570 | 0.470 |
| Phi-3.5-mini-instruct | 0.600 | 0.528 | 0.570 | 0.501 | 0.527 | 0.499 |
| **Vision-Language Models** | | | | | | |
| GPT-5.4-mini (with image inputs) | **0.936** | **0.933** | **0.901** | **0.901** | **0.94** | **0.949** |
| Gemini-3-flash (with image inputs) | 0.891 | 0.888 | 0.868 | 0.863 | 0.813 | 0.877 |
| GPT-4o (with image inputs) | 0.924 | 0.922 | 0.875 | 0.873 | 0.930 | 0.912 |
| Gemini-2.0-Flash (with image inputs) | 0.822 | 0.812 | 0.780 | 0.750 | 0.735 | 0.731 |
| Qwen2.5-VL-7B-Instruct | 0.718 | 0.714 | 0.733 | 0.727 | 0.738 | 0.733 |
| Phi-3.5-vision-instruct | 0.680 | 0.673 | 0.682 | 0.679 | 0.637 | 0.615 |
| **Language Models for Time Series** | | | | | | |
| ChatTime-7B | 0.247 | 0.242 | 0.248 | 0.247 | 0.266 | 0.261 |
| ChatTS-14B | **0.803** | **0.779** | **0.774** | **0.771** | **0.833** | **0.832** |

Table 9 presents Recognition results on the TaxoSynth dataset, which features synthetically generated sequences of varying lengths designed to reflect a taxonomy of distinct time series behaviors. Performance patterns largely mirror those seen in SUSHI, with GPT-5.4-mini (with image input) achieving the strongest scores across all distractor types. ChatTS continues to perform competitively, while GPT-5.4-mini remains the strongest language-only model, reaffirming the value of instruction tuning and visual input on complex, structured sequence data.

*Table 10.* Performance of Language and Vision-Language Models on NICU-HR Dataset in the Recognition Setting using 4 Different Negative Sampling Techniques

| Model | S-Bert Embeddings with Cosine Similarity | | DTW Distance | | Euclidean Distance | | Longest Common Subsequence | |
|---|---|---|---|---|---|---|---|---|
| | Accuracy | F1 Score | Accuracy | F1 Score | Accuracy | F1 Score | Accuracy | F1 Score |
| **Language Models** | | | | | | | | |
| GPT-5.4-mini | **0.681** | **0.678** | **0.629** | **0.628** | **0.626** | **0.624** | **0.723** | **0.722** |
| Gemini-3-flash | 0.680 | 0.677 | 0.621 | 0.618 | 0.609 | 0.603 | 0.679 | 0.672 |
| GPT-4o | 0.669 | 0.665 | 0.567 | 0.563 | 0.544 | 0.543 | 0.681 | 0.677 |
| GPT-4o (LLMTime) | 0.671 | 0.668 | 0.569 | 0.565 | 0.618 | 0.617 | 0.721 | 0.721 |
| Gemini-2.0-Flash | 0.643 | 0.642 | 0.557 | 0.557 | 0.541 | 0.540 | 0.663 | 0.661 |
| Llama-3.1-8B-Instruct | 0.503 | 0.502 | 0.458 | 0.454 | 0.515 | 0.513 | 0.521 | 0.518 |
| Qwen2.5-14B-Instruct-1M | 0.623 | 0.619 | 0.540 | 0.541 | 0.523 | 0.522 | 0.588 | 0.586 |
| Qwen2.5-7B-Instruct-1M | 0.647 | 0.646 | 0.530 | 0.527 | 0.518 | 0.513 | 0.574 | 0.573 |
| Phi-3.5-mini-instruct | 0.540 | 0.541 | 0.506 | 0.505 | 0.494 | 0.493 | 0.538 | 0.533 |
| **Vision-Language Models** | | | | | | | | |
| GPT-5.4-mini (with image inputs) | **0.698** | **0.692** | **0.694** | **0.692** | **0.672** | **0.668** | **0.782** | **0.778** |
| Gemini-3-flash (with image inputs) | 0.674 | 0.671 | 0.658 | 0.654 | 0.601 | 0.598 | 0.691 | 0.688 |
| GPT-4o (with image inputs) | 0.671 | 0.670 | 0.604 | 0.602 | 0.657 | 0.654 | 0.749 | 0.746 |
| Gemini-2.0-Flash (with image inputs) | 0.657 | 0.654 | 0.597 | 0.594 | 0.589 | 0.588 | 0.679 | 0.674 |
| Qwen2.5-VL-7B-Instruct | 0.614 | 0.610 | 0.544 | 0.542 | 0.538 | 0.535 | 0.621 | 0.617 |
| Phi-3.5-vision-instruct | 0.551 | 0.551 | 0.529 | 0.527 | 0.503 | 0.501 | 0.543 | 0.539 |
| **Language Models for Time Series** | | | | | | | | |
| ChatTime-7B | 0.624 | 0.621 | 0.543 | 0.542 | 0.454 | 0.453 | 0.431 | 0.429 |
| ChatTS-14B | **0.638** | **0.633** | **0.545** | **0.545** | **0.532** | **0.531** | **0.601** | **0.597** |

Table 10 presents Recognition results on the NICU-HR dataset. GPT-5.4-mini text and vision variants consistently achieve the best performance across all negative sampling methods. Gemini-3-Flash variants perform competitively but consistently below GPT-5.4-mini, while Llama and Phi-mini models represent the weakest performers, showing limited capacity for smaller open-weights models. VLMs generally outperform pure language models, suggesting that explicit visual representations of time series significantly improve recognition performance. TSLMs (ChatTS, ChatTime) outperform weaker general LMs but still lag behind top VLMs. Finally, model rankings hold across all negative sampling strategies.

*Table 11.* Performance of Various Language and Vision-Language Models on TRUCE-Stock Dataset in the Differentiation Setting using 4 Different Negative Sampling Techniques

| Model | S-Bert Embeddings with Cosine Similarity | | DTW Distance | | Euclidean Distance | | Longest Common Subsequence | |
|---|---|---|---|---|---|---|---|---|
| | Accuracy | F1 Score | Accuracy | F1 Score | Accuracy | F1 Score | Accuracy | F1 Score |
| **Language Models** | | | | | | | | |
| GPT-5.4-mini | **0.611** | **0.607** | **0.608** | **0.606** | **0.603** | **0.594** | **0.628** | **0.624** |
| Gemini-3-flash | 0.522 | 0.521 | 0.581 | 0.580 | 0.528 | 0.522 | 0.562 | 0.562 |
| GPT-4o | 0.531 | 0.531 | 0.524 | 0.523 | 0.514 | 0.514 | 0.566 | 0.565 |
| GPT-4o (LLMTime) | 0.584 | 0.574 | 0.562 | 0.557 | 0.543 | 0.543 | 0.620 | 0.590 |
| Gemini-2.0-Flash | 0.505 | 0.501 | 0.529 | 0.528 | 0.518 | 0.517 | 0.499 | 0.497 |
| Llama-3.1-8B-Instruct | 0.310 | 0.261 | 0.259 | 0.195 | 0.242 | 0.183 | 0.249 | 0.182 |
| Qwen2.5-14B-Instruct-1M | 0.471 | 0.470 | 0.534 | 0.533 | 0.521 | 0.520 | 0.431 | 0.429 |
| Qwen2.5-7B-Instruct-1M | 0.474 | 0.465 | 0.429 | 0.414 | 0.438 | 0.422 | 0.290 | 0.265 |
| Phi-3.5-mini-instruct | 0.468 | 0.467 | 0.320 | 0.322 | 0.318 | 0.319 | 0.319 | 0.322 |
| **Vision-Language Models** | | | | | | | | |
| GPT-5.4-mini (with image inputs) | **0.645** | **0.644** | **0.733** | **0.730** | **0.676** | **0.676** | **0.643** | **0.640** |
| Gemini-3-flash (with image inputs) | 0.628 | 0.622 | 0.684 | 0.681 | 0.618 | 0.611 | 0.549 | 0.545 |
| GPT-4o (with image inputs) | 0.631 | 0.630 | 0.669 | 0.669 | 0.657 | 0.657 | 0.541 | 0.541 |
| Gemini-2.0-Flash (with image inputs) | 0.608 | 0.592 | 0.650 | 0.607 | 0.604 | 0.600 | 0.501 | 0.500 |
| Qwen2.5-VL-7B-Instruct | 0.526 | 0.525 | 0.623 | 0.624 | 0.621 | 0.621 | 0.481 | 0.481 |
| Phi-3.5-vision-instruct | 0.589 | 0.588 | 0.612 | 0.613 | 0.603 | 0.604 | 0.577 | 0.559 |
| **Language Models for Time Series** | | | | | | | | |
| ChatTime-7B | 0.040 | 0.038 | 0.074 | 0.071 | 0.072 | 0.069 | 0.048 | 0.045 |
| ChatTS-14B | **0.492** | **0.492** | **0.412** | **0.409** | **0.393** | **0.389** | **0.501** | **0.499** |

Table 11 shows Differentiation task performance on the TRUCE-Stock dataset. As in the Recognition setting, this real-world dataset proves difficult across all model classes, with notably lower F1 scores. GPT-5.4-mini again leads among LLMs, and GPT-5.4-mini with image input remains strongest overall, though margins are narrower than in Recognition. ChatTS continues to outperform most open-source LLMs, including its base LLM, reinforcing its strength in fine-grained comparison tasks under realistic conditions, even on the more difficult Differentiation setting.

*Table 12.* Performance of Various Language and Vision-Language Models on TRUCE-Synthetic Dataset in the Differentiation Setting using 4 Different Negative Sampling Techniques

| Model | S-Bert Embeddings with Cosine Similarity | | DTW Distance | | Euclidean Distance | | Longest Common Subsequence | |
|---|---|---|---|---|---|---|---|---|
| | Accuracy | F1 Score | Accuracy | F1 Score | Accuracy | F1 Score | Accuracy | F1 Score |
| **Language Models** | | | | | | | | |
| GPT-5.4-mini | **0.832** | **0.817** | **0.844** | **0.841** | **0.811** | **0.810** | **0.789** | **0.784** |
| Gemini-3-flash | 0.756 | 0.755 | 0.754 | 0.753 | 0.688 | 0.683 | 0.774 | 0.771 |
| GPT-4o | 0.751 | 0.752 | 0.725 | 0.724 | 0.696 | 0.696 | 0.692 | 0.692 |
| GPT-4o (LLMTime) | 0.819 | 0.809 | 0.794 | 0.794 | 0.804 | 0.809 | 0.787 | 0.786 |
| Gemini-2.0-Flash | 0.745 | 0.741 | 0.629 | 0.625 | 0.592 | 0.591 | 0.690 | 0.665 |
| Llama-3.1-8B-Instruct | 0.342 | 0.284 | 0.275 | 0.217 | 0.267 | 0.210 | 0.257 | 0.200 |
| Qwen2.5-14B-Instruct-1M | 0.720 | 0.721 | 0.681 | 0.683 | 0.656 | 0.656 | 0.616 | 0.617 |
| Qwen2.5-7B-Instruct-1M | 0.612 | 0.607 | 0.470 | 0.463 | 0.445 | 0.433 | 0.417 | 0.402 |
| Phi-3.5-mini-instruct | 0.618 | 0.619 | 0.373 | 0.379 | 0.369 | 0.369 | 0.346 | 0.347 |
| **Vision-Language Models** | | | | | | | | |
| GPT-5.4-mini (with image inputs) | **0.844** | **0.843** | **0.881** | **0.876** | **0.818** | **0.817** | **0.792** | **0.787** |
| Gemini-3-flash (with image inputs) | 0.811 | 0.808 | 0.834 | 0.832 | 0.786 | 0.782 | 0.766 | 0.762 |
| GPT-4o (with image inputs) | 0.801 | 0.801 | 0.808 | 0.807 | 0.769 | 0.769 | 0.729 | 0.728 |
| Gemini-2.0-Flash (with image inputs) | 0.789 | 0.786 | 0.745 | 0.739 | 0.727 | 0.725 | 0.704 | 0.700 |
| Qwen2.5-VL-7B-Instruct | 0.754 | 0.754 | 0.732 | 0.732 | 0.719 | 0.718 | 0.665 | 0.666 |
| Phi-3.5-vision-instruct | 0.802 | 0.802 | 0.708 | 0.709 | 0.694 | 0.693 | 0.657 | 0.658 |
| **Language Models for Time Series** | | | | | | | | |
| ChatTime-7B | 0.130 | 0.127 | 0.104 | 0.101 | 0.109 | 0.107 | 0.134 | 0.134 |
| ChatTS-14B | **0.654** | **0.652** | **0.593** | **0.588** | **0.601** | **0.598** | **0.711** | **0.708** |

Table 12 presents Differentiation performance on the TRUCE-Synthetic dataset. Overall scores are slightly lower than in the Recognition setting, reflecting the added complexity of multi-choice reasoning. GPT-5.4-mini language and vision variants lead across all sampling strategies, while ChatTS again stands out among TSLMs, outperforming many open-source LLMs, including its base LLM. This pattern underscores the benefit of both vision-based inputs and domain-specific training for tasks requiring contrastive reasoning.

*Table 13.* Performance of Various Language and Vision-Language Models on SUSHI Dataset in the Differentiation Setting using 4 Different Negative Sampling Techniques

| Model | S-Bert Embeddings with Cosine Similarity | | DTW Distance | | Euclidean Distance | | Longest Common Subsequence | |
|---|---|---|---|---|---|---|---|---|
| | Accuracy | F1 Score | Accuracy | F1 Score | Accuracy | F1 Score | Accuracy | F1 Score |
| **Language Models** | | | | | | | | |
| GPT-5.4-mini | **0.854** | **0.851** | **0.763** | **0.761** | **0.679** | **0.677** | **0.979** | **0.978** |
| Gemini-3-flash | 0.811 | 0.808 | 0.722 | 0.722 | 0.627 | 0.624 | 0.918 | 0.912 |
| GPT-4o | 0.802 | 0.803 | 0.646 | 0.648 | 0.583 | 0.584 | 0.916 | 0.916 |
| GPT-4o (LLMTime) | 0.850 | 0.823 | 0.673 | 0.670 | 0.615 | 0.614 | 0.975 | 0.959 |
| Gemini-2.0-Flash | 0.728 | 0.711 | 0.625 | 0.549 | 0.527 | 0.488 | 0.880 | 0.827 |
| Llama-3.1-8B-Instruct | 0.385 | 0.382 | 0.226 | 0.214 | 0.239 | 0.221 | 0.380 | 0.376 |
| Qwen2.5-14B-Instruct-1M | 0.740 | 0.738 | 0.446 | 0.431 | 0.365 | 0.351 | 0.792 | 0.792 |
| Qwen2.5-7B-Instruct-1M | 0.636 | 0.633 | 0.239 | 0.216 | 0.178 | 0.150 | 0.557 | 0.540 |
| Phi-3.5-mini-instruct | 0.643 | 0.648 | 0.444 | 0.443 | 0.344 | 0.344 | 0.716 | 0.721 |
| **Vision-Language Models** | | | | | | | | |
| GPT-5.4-mini (with image inputs) | **0.988** | **0.982** | **0.981** | **0.978** | **0.966** | **0.963** | **0.992** | **0.988** |
| Gemini-3-flash (with image inputs) | 0.978 | 0.977 | 0.937 | 0.932 | 0.911 | 0.906 | 0.984 | 0.978 |
| GPT-4o (with image inputs) | 0.981 | 0.981 | 0.965 | 0.965 | 0.951 | 0.951 | 0.988 | 0.988 |
| Gemini-2.0-Flash (with image inputs) | 0.976 | 0.969 | 0.896 | 0.894 | 0.898 | 0.893 | 0.972 | 0.940 |
| Qwen2.5-VL-7B-Instruct | 0.974 | 0.974 | 0.871 | 0.871 | 0.851 | 0.851 | 0.968 | 0.968 |
| Phi-3.5-vision-instruct | 0.808 | 0.807 | 0.756 | 0.755 | 0.716 | 0.713 | 0.864 | 0.865 |
| **Language Models for Time Series** | | | | | | | | |
| ChatTime-7B | 0.287 | 0.285 | 0.235 | 0.231 | 0.261 | 0.260 | 0.281 | 0.283 |
| ChatTS-14B | **0.783** | **0.781** | **0.602** | **0.595** | **0.574** | **0.570** | **0.809** | **0.808** |

Table 13 summarizes Differentiation results on the SUSHI dataset. GPT-5.4-mini with image input achieves near-perfect F1 scores across all distractor types, followed closely by other VLMs, confirming the advantage of visual modality in structured settings. GPT-5.4-mini continues to lead among text-only models, while ChatTS again proves competitive, performing on par with or better than many open-source LLMs, including its base LLM.

*Table 14.* Performance of Various Language and Vision-Language Models on TaxoSynth Dataset in the Differentiation Setting using 3 Different Negative Sampling Techniques

| Model | S-Bert Embeddings with Cosine Similarity | | DTW Distance | | Longest Common Subsequence | |
|---|---|---|---|---|---|---|
| | Accuracy | F1 Score | Accuracy | F1 Score | Accuracy | F1 Score |
| **Language Models** | | | | | | |
| GPT-5.4-mini | **0.812** | **0.806** | **0.765** | **0.763** | **0.723** | **0.717** |
| Gemini-3-flash | 0.745 | 0.739 | 0.678 | 0.668 | 0.701 | 0.694 |
| GPT-4o | 0.716 | 0.716 | 0.709 | 0.710 | 0.655 | 0.655 |
| GPT-4o (LLMTime) | 0.794 | 0.781 | 0.752 | 0.747 | 0.696 | 0.687 |
| Gemini-2.0-Flash | 0.642 | 0.639 | 0.607 | 0.602 | 0.654 | 0.633 |
| Llama-3.1-8B-Instruct | 0.394 | 0.283 | 0.259 | 0.254 | 0.287 | 0.282 |
| Qwen2.5-14B-Instruct-1M | 0.668 | 0.667 | 0.527 | 0.520 | 0.658 | 0.658 |
| Qwen2.5-7B-Instruct-1M | 0.589 | 0.584 | 0.349 | 0.344 | 0.455 | 0.447 |
| Phi-3.5-mini-instruct | 0.596 | 0.593 | 0.495 | 0.497 | 0.524 | 0.528 |
| **Vision-Language Models** | | | | | | |
| GPT-5.4-mini (with image inputs) | **0.823** | **0.826** | **0.842** | **0.838** | **0.827** | **0.824** |
| Gemini-3-flash (with image inputs) | 0.814 | 0.811 | 0.773 | 0.770 | 0.772 | 0.766 |
| GPT-4o (with image inputs) | 0.805 | 0.805 | 0.812 | 0.812 | 0.797 | 0.797 |
| Gemini-2.0-Flash (with image inputs) | 0.707 | 0.706 | 0.731 | 0.725 | 0.745 | 0.732 |
| Qwen2.5-VL-7B-Instruct | 0.767 | 0.763 | 0.759 | 0.753 | 0.748 | 0.748 |
| Phi-3.5-vision-instruct | 0.719 | 0.700 | 0.689 | 0.686 | 0.708 | 0.712 |
| **Language Models for Time Series** | | | | | | |
| ChatTime-7B | 0.175 | 0.172 | 0.150 | 0.147 | 0.176 | 0.176 |
| ChatTS-14B | **0.756** | **0.754** | **0.650** | **0.647** | **0.786** | **0.785** |

Table 14 reports Differentiation results on the TaxoSynth dataset. GPT-5.4-mini vision and text perform best overall, with consistent results across all sampling strategies. ChatTS achieves strong performance, narrowing the gap with proprietary models and outperforming its base LLM, showing robustness even in longer, more complex multi-choice tasks.

*Table 15.* Performance of Language and Vision-Language Models on NICU-HR Dataset in the Differentiation Setting using 4 Different Negative Sampling Techniques

| Model | S-Bert Embeddings with Cosine Similarity | | DTW Distance | | Euclidean Distance | | Longest Common Subsequence | |
|---|---|---|---|---|---|---|---|---|
| | Accuracy | F1 Score | Accuracy | F1 Score | Accuracy | F1 Score | Accuracy | F1 Score |
| **Language Models** | | | | | | | | |
| GPT-5.4-mini | **0.916** | **0.901** | **0.965** | **0.963** | **0.954** | **0.952** | **0.948** | **0.945** |
| Gemini-3-flash | 0.891 | 0.883 | 0.952 | 0.946 | 0.951 | 0.949 | 0.944 | 0.944 |
| GPT-4o | 0.865 | 0.862 | 0.944 | 0.942 | 0.945 | 0.941 | 0.938 | 0.932 |
| GPT-4o (LLMTime) | 0.871 | 0.870 | 0.948 | 0. 945 | 0.949 | 0.949 | 0.941 | 0.939 |
| Gemini-2.0-Flash | 0.854 | 0.851 | 0.938 | 0.932 | 0.943 | 0.943 | 0.930 | 0.927 |
| Llama-3.1-8B-Instruct | 0.588 | 0.583 | 0.689 | 0.687 | 0.645 | 0.644 | 0.691 | 0.688 |
| Qwen2.5-14B-Instruct-1M | 0.743 | 0.743 | 0.814 | 0.811 | 0.724 | 0.721 | 0.779 | 0.778 |
| Qwen2.5-7B-Instruct-1M | 0.694 | 0.692 | 0.782 | 0.780 | 0.726 | 0.721 | 0.789 | 0.788 |
| Phi-3.5-mini-instruct | 0.678 | 0.677 | 0.765 | 0.761 | 0.694 | 0.688 | 0.734 | 0.731 |
| **Vision-Language Models** | | | | | | | | |
| GPT-5.4-mini (with image inputs) | **0.924** | **0.918** | **0.976** | **0.973** | **0.963** | **0.963** | **0.966** | **0.961** |
| Gemini-3-flash (with image inputs) | 0.901 | 0.897 | 0.959 | 0.952 | 0.952 | 0.946 | 0.949 | 0.947 |
| GPT-4o (with image inputs) | 0.887 | 0.882 | 0.957 | 0.954 | 0.917 | 0.912 | 0.899 | 0.896 |
| Gemini-2.0-Flash (with image inputs) | 0.869 | 0.865 | 0.943 | 0.941 | 0.946 | 0.944 | 0.939 | 0.938 |
| Qwen2.5-VL-7B-Instruct | 0.797 | 0.796 | 0.793 | 0.792 | 0.788 | 0.782 | 0.791 | 0.787 |
| Phi-3.5-vision-instruct | 0.703 | 0.702 | 0.771 | 0.771 | 0.708 | 0.701 | 0.744 | 0.741 |
| **Language Models for Time Series** | | | | | | | | |
| ChatTime-7B | 0.629 | 0.622 | 0.727 | 0.723 | 0.673 | 0.669 | 0.618 | 0.615 |
| ChatTS-14B | **0.786** | **0.783** | **0.821** | **0.819** | **0.783** | **0.783** | **0.818** | **0.812** |

Table 15 reports Differentiation results on the NICU-HR dataset. GPT-5.4-mini vision and text variants consistently achieve the strongest performance, followed closely by Gemini variants, while Llama-3.1-8B and smaller Phi-mini models perform worse. Vision-language models generally match or slightly outperform strong language models across settings. ChatTS clearly outperforms ChatTime and smaller LLMs but remains below comparably sized vision-language models. The ranking of models is highly stable across all negative sampling strategies, indicating that model capability, rather than the choice of sampling method, primarily drives performance differences. Overall, larger and multimodal models consistently lead, while smaller language only models lag behind.

These results offer fine-grained insight into model behavior and reinforce that our main findings are robust across contrastive construction strategies. (1) Across datasets, we find that vision-language models consistently outperform their text-only counterparts. (2) This performance gap is most evident in synthetic datasets like SUSHI and TRUCE-Synthetic and real-world NICU-HR data, where models such as GPT-4o with image input achieve near-perfect F1 scores. (3) In contrast, the real-world shorter and noisier TRUCE-Stock dataset presents a more substantial challenge, with significantly lower performance across all models, including proprietary LLMs and VLMs. (4) While vision-based models lead in most settings, strong time-series-specific language models like ChatTS perform competitively in the Differentiation task, especially on datasets such as TaxoSynth where they outperform many open-source and some proprietary LLMs. (6) BPE-based prompting (as in GPT-4o + LLMTime) has a more pronounced effect on performance than model scale alone, and consistently improves robustness across both tasks. (7) Finally, model rankings remain remarkably stable across all datasets, contrastive sampling strategies, and task settings, underscoring that the observed trends are not artifacts of any specific evaluation construction. This consistency holds across synthetic and real-world datasets, across short and long time series, and across both simple and complex descriptions. The fact that model ordering is preserved despite this breadth of variation demonstrates the robustness and diagnostic reliability of the BEDTime.

*Note:* Euclidean distance was excluded for TaxoSynth due to its variable-length sequences, which prevent direct computation.

Figure 5 presents a side-by-side comparison of LLM and VLM performance across all datasets, showing that VLM variants consistently outperform their corresponding LLM counterparts. Since VLMs share the same underlying language models, these gains isolate the contribution of visual encoders, indicating that explicit visual representations of time-series structure improve performance.

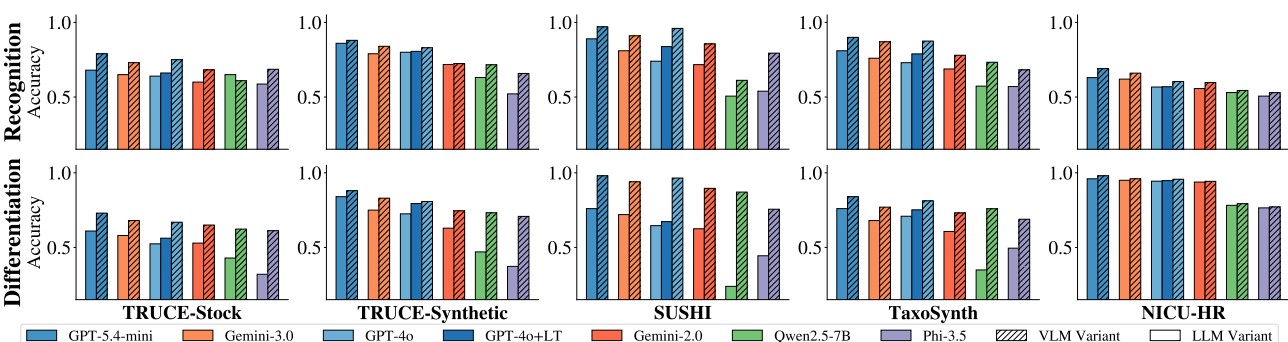

*Figure 5.* Accuracy of LLMs and VLMs on recognition and differentiation tasks across real-world (TRUCE-Stock, NICU-HR) and synthetic (TRUCE-Synthetic, SUSHI, TaxoSynth) time series datasets. Negative samples for contrastive evaluation were generated using Dynamic Time Warping (DTW). The consistent performance gains of VLMs, especially on the differentiation task, highlight the importance of visual cues for robust time series analysis.

# F. Per-Class Accuracy Analysis

To better understand why models struggle on certain time series, we compute per-class accuracy on SUSHI for both Recognition and Differentiation, stratifying the dataset into 7 signal types of increasing structural complexity: clean, smooth, negative spike (n_spike), positive spike (p_spike), step-wise, positive-and-negative spike (pn_spike), and noisy. Full results are reported in Table 16.

All modalities achieve near-perfect accuracy on the simplest signal classes (clean, smooth), confirming that every model can handle basic structural patterns. However, performance degrades consistently as signal complexity increases, following a clear hierarchy: negative spike ≈ positive spike ≈ step-wise > positive-and-negative spike > noisy. This degradation is most pronounced for LLMs—smaller open-weight LLMs such as Llama-3.1-8B and Qwen2.5-7B fall below random baseline on noisy signals for Differentiation (0.01 and 0.09 respectively), indicating a near-complete failure to distinguish descriptions when the underlying time series contains significant noise. Even among larger LLMs, the drop from clean to noisy is substantial: GPT-4o drops from 0.98 to 0.49 on Recognition and from 0.82 to 0.38 on Differentiation. The addition of frontier models GPT-5.4-mini and Gemini-3.0 improves robustness to complexity—GPT-5.4-mini maintains 0.78 on noisy Recognition—but the degradation pattern persists.

VLMs, by contrast, remain remarkably robust across all complexity levels. GPT-4o-Vision maintains 0.92 on noisy Recognition and 0.95 on noisy Differentiation, while GPT-5.4-mini-Vision achieves 0.94 and 0.97 respectively. Even the smallest VLM, Qwen2.5-VL-7B, substantially outperforms comparably-sized LLMs on complex classes, demonstrating that visual encoders provide consistent advantages when structural patterns become harder to detect from numeric representations alone.

TSLMs show an intermediate pattern: ChatTS-14B degrades more slowly than LLMs of comparable or larger size, dropping from 0.98 to 0.57 on Recognition (compared to GPT-4o's 0.98 to 0.49). This confirms that direct numeric encoding provides robustness over text tokenization of numeric values, but does not capture the full benefit of visual representation. ChatTime-7B, however, performs poorly across all classes due to its high refusal rates, as discussed in Appendix I.

These per-class trends are consistent with the robustness findings reported in Section 4.2.3, where we observe similar modality-dependent sensitivity to perturbations such as noise injection and temporal shifting.

# G. Complete Robustness Test Results

Figure 6 summarizes the complete results across both recognition and differentiation tasks under five real-world perturbations. Across perturbations, both tasks exhibit broadly similar trends: for linear interpolation, as the time series length increases, LLM performance degrades, whereas TSLMs remain more robust to length increases and, in some cases, benefit from the increased signal density induced by interpolation. As missing data increases, LLM performance steadily declines, with a sharp drop observed once missingness exceeds 25%, affecting both tasks. For amplitude scaling, both LLMs and TSLMs appear largely robust, with no major degradation in performance. Under added Gaussian noise, all models degrade as noise increases, but LLMs show the steepest decline, followed by TSLMs, while VLMs degrade more gradually, indicating

*Table 16.* Per-class Recognition (R) and Differentiation (D) accuracy on SUSHI using DTW distance for negative sampling. Signal types are ordered by increasing structural complexity from left to right: clean, smooth, negative spike (n_spike), positive spike (p_spike), step-wise, positive-and-negative spike (pn_spike), and noisy. All modalities achieve near-perfect accuracy on simple classes but degrade with complexity, with LLMs most affected and VLMs remaining robust.

| | Model | Recognition (R) | | | | | | | Differentiation (D) | | | | | | |
|---|---|---|---|---|---|---|---|---|---|---|---|---|---|---|---|
| | | Clean | Smooth | N_Spike | P_Spike | Step | PN_Spike | Noisy | Clean | Smooth | N_Spike | P_Spike | Step | PN_Spike | Noisy |
| LLMs | GPT-5.4-mini | .99 | .98 | .88 | .90 | .89 | .81 | .78 | .96 | .93 | .79 | .80 | .74 | .61 | .49 |
| | Gemini-3.0 | .99 | .96 | .83 | .80 | .81 | .66 | .63 | .92 | .89 | .76 | .75 | .70 | .57 | .45 |
| | GPT-4o | .98 | .94 | .73 | .77 | .74 | .54 | .49 | .82 | .85 | .69 | .68 | .63 | .50 | .38 |
| | GPT-4o+LT | .99 | .98 | .85 | .86 | .85 | .69 | .68 | .87 | .84 | .71 | .70 | .65 | .52 | .40 |
| | Gemini-2.0 | .92 | .95 | .75 | .71 | .72 | .53 | .48 | .80 | .83 | .67 | .66 | .61 | .48 | .36 |
| | Qwen2.5-14B | .68 | .65 | .50 | .53 | .51 | .37 | .34 | .62 | .65 | .48 | .49 | .43 | .30 | .18 |
| | Qwen2.5-7B | .68 | .65 | .53 | .50 | .51 | .37 | .34 | .42 | .39 | .27 | .28 | .22 | .02 | .09 |
| | Llama-3.1-8B | .65 | .62 | .51 | .48 | .49 | .36 | .32 | .40 | .37 | .26 | .26 | .21 | .09 | .01 |
| | Phi-3.5 | .69 | .72 | .56 | .53 | .54 | .36 | .39 | .61 | .64 | .47 | .48 | .42 | .29 | .17 |
| VLMs | GPT-5.4-mini | .99 | .99 | .98 | .97 | .95 | .97 | .94 | .99 | .99 | .99 | .98 | .99 | .95 | .97 |
| | Gemini-3.0 | .99 | .98 | .91 | .92 | .91 | .82 | .84 | .99 | .99 | .95 | .96 | .93 | .90 | .86 |
| | GPT-4o | .99 | .99 | .97 | .97 | .93 | .96 | .92 | .99 | .99 | .98 | .99 | .97 | .93 | .95 |
| | Gemini-2.0 | .99 | .97 | .85 | .88 | .86 | .73 | .75 | .99 | .98 | .92 | .92 | .89 | .83 | .77 |
| | Qwen2.5-VL-7B | .81 | .78 | .60 | .64 | .61 | .45 | .40 | .99 | .98 | .90 | .90 | .86 | .77 | .69 |
| | Phi-3.5-vision | .95 | .98 | .78 | .81 | .79 | .63 | .59 | .96 | .93 | .80 | .79 | .74 | .61 | .49 |
| TSLMs | ChatTS-14B | .98 | .95 | .77 | .81 | .78 | .61 | .57 | .80 | .77 | .63 | .64 | .58 | .45 | .33 |
| | ChatTime-7B | .37 | .36 | .27 | .29 | .28 | .20 | .19 | .42 | .39 | .27 | .28 | .22 | .10 | .01 |

comparatively higher robustness to noise. Finally, under image quality degradation, VLM performance consistently decreases as pixelation increases, and these trends hold across both recognition and differentiation tasks.

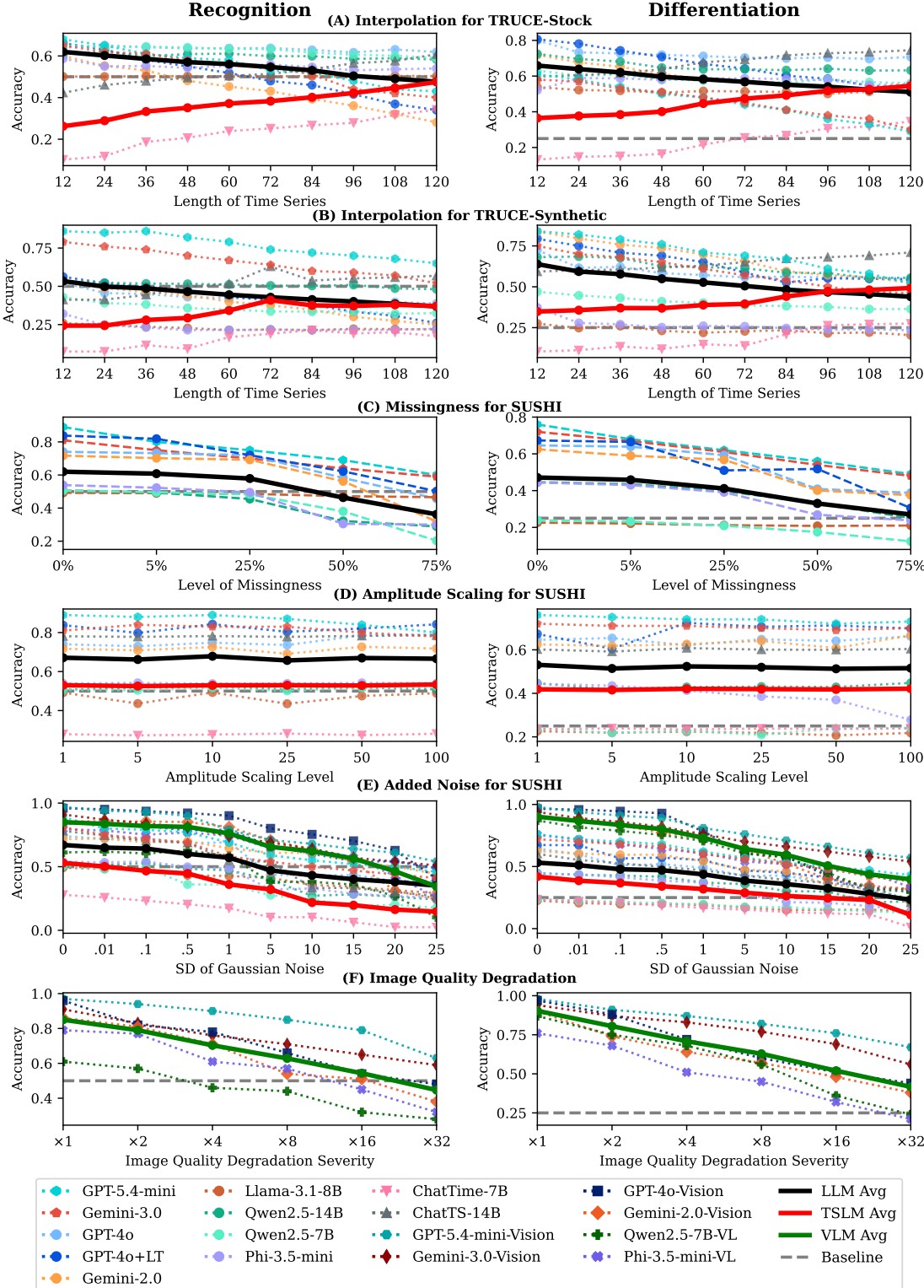

*Figure 6.* Robustness of LMs to **five** realistic perturbations for both Recognition and Differentiation. Accuracy decline for LLMs as time series grow longer or contain more missing values. As signals get noisier performance decline. LMs are relatively robust to amplitude scaling. For VLMs, accuracy decreases as plot resolution degrades. DTW distance is used to pick distractors.

## H. Implementation Details

Experiments are run through the OpenAI GPT-4o and GPT-5.4-mini API and the Google Gemini-2.0-flash and Gemini-3-flash-preview API and model inference endpoints on Hugging face for Qwen, Phi, Llama, ChatTS and ChatTime models. Batching is used where possible to minimize API overhead. Inference is parallelized across NVIDIA A6000 GPUs for models requiring local deployment. Each experiment is repeated with all distractor sampling methods to ensure robustness of results.

## I. Why does ChatTime perform worse than random guessing on recognition and differentiation tasks?

Upon analysis, we found that ChatTime often fails to follow the task instructions, which significantly affects its performance in these classification-based tasks. Specifically, rather than outputting a discrete label (e.g., "A/B/C/D" or "True/False"), ChatTime frequently returns a numeric value extracted directly from the input time series. Since our evaluation requires a well-formed answer to compare against the gold label, any non-conforming output is marked as incorrect, which brings down both accuracy and F1 considerably. To better understand this behavior, we tracked refusal rates, the percentage of prompts where ChatTime did not produce a valid classification. As shown in Table below, these refusal rates are very high, especially in recognition tasks. We hypothesize that ChatTime's high error rates on instruction-following tasks like recognition and differentiation stem from its architectural and training design. While ChatTime incorporates instruction fine-tuning, its base LLM is Llama-2-7B-Base, a non-instruction tuned model that undergoes continuous pretraining on time series data before instruction tuning. The fine-tuning phase focuses on generation-based tasks such as forecasting and time series QA, but not on multiple-choice classification with strict formatting constraints. As a result, ChatTime often returns only numeric outputs rather than discrete classification labels (e.g., "A", or "True"), especially when the prompt requires strict instruction-following behavior. Table17 - Refusal rates for both Recognition and differentiation tasks across all five datasets, using DTW-based negative sampling.

*Table 17.* Refusal rates across recognition and differentiation tasks on different datasets.

| Task | Dataset | Refusal Rate (%) |
|---|---|---|
| **Recognition** | TRUCE-Stock | 83.4 |
| | TRUCE-Synthetic | 79.1 |
| | SUSHI | 61.6 |
| | TaxoSynth | 64.3 |
| | NICU-HR | 36.0 |
| **Differentiation** | TRUCE-Stock | 74.2 |
| | TRUCE-Synthetic | 67.0 |
| | SUSHI | 53.5 |
| | TaxoSynth | 58.7 |
| | NICU-HR | 8.1 |

## J. Additional Baseline Metrics

We ran additional Baseline comparison metrics against the generated descriptions by the top performer LLM, VLM and TSLM and the ground truth on SUSHI dataset: **Average Edit Distance:** GPT-4o Vision: 473.15, ChatTS: 426.22,GPT-4o: 434.68. **Average Token Overlap:** GPT-4o Vision: 0.071, ChatTS: 0.068, GPT-4o: 0.067.

The low token overlap and moderate edit distances suggest that exact matching metrics (such as BLEU or token overlap) may not fully capture quality in this task. These results support our choice to favor NLI-based entailment over string-based heuristics such as token overlap or edit distance.

Additionally, we recomputed the NLI metrics for the testing samples that include no numbers. This provides a result on samples where numeric alterations cannot be a problem. Our results are in Table 18 below, where "w/ #" denotes the dataset with all samples (N=1400), including those with numbers as in the paper, and "w/o #" denotes the dataset without numbers (N=1279). Here we see that there is minimal difference between the scores for datasets with and without numbers,

suggesting numeric misalignment have minimal impact.

*Table 18.* DeBERTa NLI entailment percentages when ground truth numeric values are included (w/#) and excluded (w/o #).

| Model | Bi-dir | Gen → GT | GT → Gen |
|---|---|---|---|
| GPT-4o-Vision (w/ #) | 14.41 | 47.65 | 15.29 |
| GPT-4o-Vision (w/o #) | 15.76 | 49.84 | 16.72 |
| GPT-4o (w/ #) | 2.94 | 9.71 | 6.18 |
| GPT-4o (w/o #) | 3.22 | 10.61 | 6.43 |
| ChatTS-14B (w/ #) | 2.65 | 23.53 | 2.65 |
| ChatTS-14B (w/o #) | 2.89 | 25.40 | 2.89 |

While the raw DeBERTa entailment percentages are low in absolute terms, this behavior is expected and does not indicate poor generation quality. DeBERTa-based NLI models are highly sensitive to lexical choice and syntactic variation, and often fail to assign entailment to semantically correct paraphrases. To further address this domain mismatch, we additionally evaluate generation quality using GPT-5 as an LLM judge, which is substantially more robust to paraphrasing and semantic variation than DeBERTa.

In this setup, for each pair consisting of a ground-truth description (GT) and a model-generated description (Gen), we explicitly prompt GPT-5 to determine whether the generation entails the ground truth, whether the ground truth entails the generation, and whether both conditions hold simultaneously (bi-directional entailment). This yields the entailment-style scores reported below, which preserve the exact same relative ranking across models:

*Table 19.* GPT-5-based entailment evaluation of generated time-series descriptions on the SUSHI dataset. The table reports the percentage of cases where the model-generated description entails the ground truth (Gen→GT), the ground truth entails the generation (GT→Gen), and where entailment holds in both directions (bi-directional).

| Model | Bi-directional (%) | Gen→GT (%) | GT→Gen (%) |
|---|---|---|---|
| GPT-4o-Vision | 47.64 | 71.43 | 51.86 |
| GPT-4o-Text | 37.93 | 58.14 | 43.43 |
| ChatTS-14B | 20.93 | 46.71 | 34.93 |

These results confirm that, despite differences in absolute scale across evaluation methods, all generation metrics considered—DeBERTa NLI, number-filtered NLI, and GPT-5-based judging—produce consistent and stable model rankings. Together, they demonstrate that our generation evaluation is robust to domain mismatch, lexical variation, and evaluator choice, and does not depend on any single metric or scoring scale.

## K. Interpolation-Scaled

Figure 7 supports the analysis in Section 4.2.3 by evaluating whether scaling interpolated values mitigates the tokenization-related degradation observed in LLMs. Performance improves for most models, except Qwen2.5-14B due to reasons previously discussed.

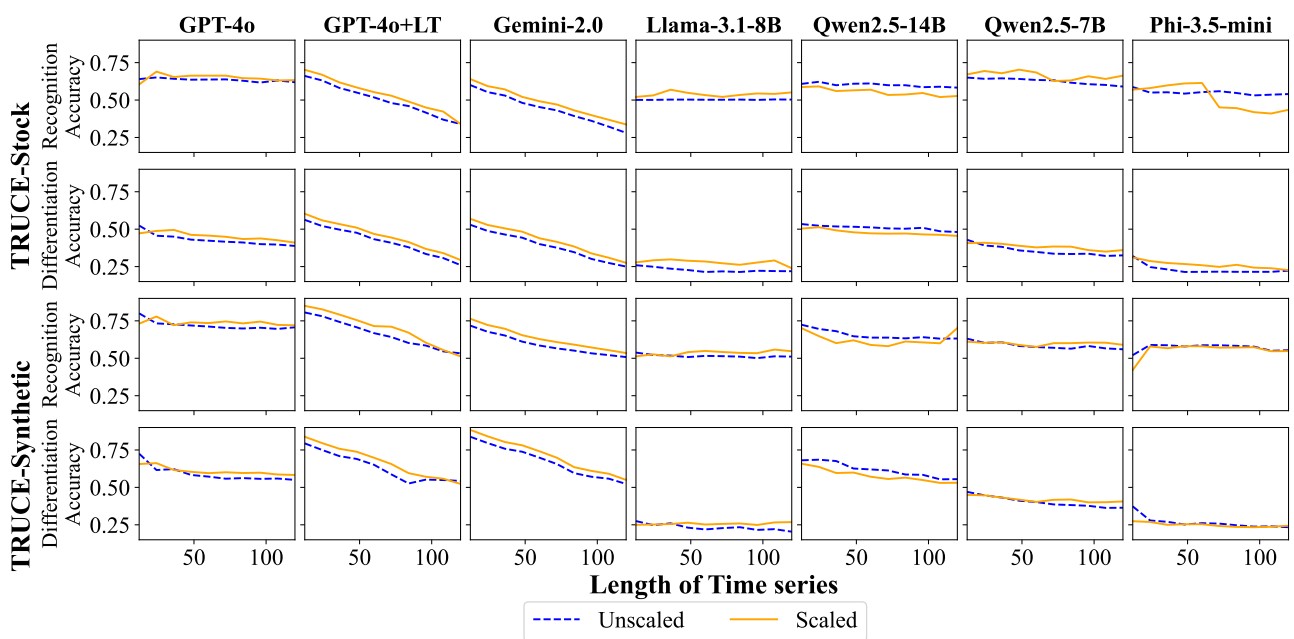

*Figure 7.* Scaling for different time series lengths

