# OpenReview forum: "BEDTime: A Unified Benchmark for Automatically Describing Time Series"
_ICML.cc/2026/Conference — ICML 2026 regular_

### Official Review · Reviewer_UCPf · 2026-03-03

**Soundness:** 3
**Presentation:** 3
**Significance:** 3
**Originality:** 3
**Overall Recommendation:** 5
**Confidence:** 5

**Summary:**

The paper introduces BEDTime, a benchmark to test whether modern multimodal time-series models can understand and describe structural properties of time series via (1) matching descriptions, (2) distinguishing correct vs. incorrect descriptions, and (3) generating free-form descriptions. Evaluating 13 models across 46,843 series from five datasets shows VLMs perform best, language-only models perform worst, specialized time-series language models underperform unexpectedly, and all models are brittle under robustness tests.

**Compliance With Llm Reviewing Policy:**

Affirmed.

**Final Justification:**

I have read the rebuttal and will keep my positive score

**Key Questions For Authors:**

1. You demonstrate that VLMs outperform dedicated TSLMs on BEDTime tasks. Do you have any hypotheses about why this is the case? Is it primarily about the visual encoders, the training data, or something else?
2. For the generation task, what are the most common failure modes you observe? Do models tend to miss certain types of structural properties more than others?

**Limitations:**

1. Bedtime only contains univariate time series dataset.

**Strengths And Weaknesses:**

S1: The paper is generally well-written and well-organized. Figure 1 provides an excellent overview, and the tables are clear and informative.
S2: The three proposed tasks (recognition, differentiation, open generation) form a logical progression from simple to complex, providing a nice spectrum for evaluating model capabilities. The negative sampling strategies appear well-thought-out.
S3: Evaluating 13 models across 5 datasets is a substantial undertaking that provides meaningful head-to-head comparisons. The inclusion of automated metrics (NLI, GPT-5) and human evaluation strengthens the results.

W1: The paper demonstrates that models fail, but provides relatively limited insight into why they fail. More qualitative analysis of common error patterns would be valuable.

---

> ### Author Rebuttal · Authors · 2026-03-31
>
> We thank the reviewer for the constructive feedback. We are encouraged that they find our work "well-written and well-organized" and recognize our evaluation as a "substantial undertaking" providing "meaningful head-to-head comparisons." We address each concern below.
>
> **W1.** The paper shows model failures but gives limited insight into why; more qualitative error analysis would help.
>
> To identify why models struggle, we computed per-class accuracy on SUSHI, stratified into 7 signal types of increasing complexity: clean, smooth, negative spike (n_spike), positive spike (p_spike), step-wise, positive-and-negative spike (pn_spike), and noisy (Table 1). We find that all modalities achieve near-perfect accuracy on simple classes (clean, smooth). Performance degrades with complexity (n_spike ≈ p_spike ≈ step-wise > pn_spike > noisy), with LLMs most affected—smaller LLMs fall below random baseline on noisy. VLMs remain consistent. TSLMs degrade slower than LLMs, confirming numeric encoding provides robustness over text tokenization but not the full benefit of visual representation. These are consistent with our robustness findings in Section 4.2.3. We incorporate these analyses in Section 4.2.
>
> Table 1. Recognition (R) and Differentiation (D) per-class accuracy for SUSHI using DTW distance for negative sampling.
>
> |Model|R-clean|R-smooth|R-n_spike|R-p_spike|R-step-wise|R-pn_spike|R-noisy|D-clean|D-smooth|D-n_spike|D-p_spike|D-step-wise|D-pn_spike|D-noisy|
> |-|-|-|-|-|-|-|-|-|-|-|-|-|-|-|
> |gpt-4o|.98|.94|.73|.77|.74|.54|.49|.82|.85|.69|.68|.63|.5|.38|
> |gpt-4o+LT|.99|.98|.85|.86|.85|.69|.68|.87|.84|.71|.7|.65|.52|.4|
> |gemini-2.0|.92|.95|.75|.71|.72|.53|.48|.8|.83|.67|.66|.61|.48|.36|
> |llama-8b|.65|.62|.51|.48|.49|.36|.32|.4|.37|.26|.26|.21|.09|.01|
> |qwen-14b|.68|.65|.5|.53|.51|.37|.34|.62|.65|.48|.49|.43|.3|.18|
> |qwen-7b|.68|.65|.53|.5|.51|.37|.34|.42|.39|.27|.28|.22|.02|.09|
> |phi-3.5|.69|.72|.56|.53|.54|.36|.39|.61|.64|.47|.48|.42|.29|.17|
> |gpt-4o-V|.99|.99|.97|.97|.93|.96|.92|.99|.99|.98|.99|.97|.93|.95|
> |gemini-2.0-V|.99|.97|.85|.88|.86|.73|.75|.99|.98|.92|.92|.89|.83|.77|
> |qwen-vl-7b|.81|.78|.6|.64|.61|.45|.4|.99|.98|.9|.9|.86|.77|.69|
> |phi-3.5-V|.95|.98|.78|.81|.79|.63|.59|.96|.93|.8|.79|.74|.61|.49|
> |chattime|.37|.36|.27|.29|.28|.2|.19|.42|.39|.27|.28|.22|.1|.01|
> |chatts|.98|.95|.77|.81|.78|.61|.57|.80|.77|.63|.64|.58|.45|.33|
> |gpt-5.4-mini|.99|.98|.88|.9|.89|.81|.78|.96|.93|.79|.8|.74|.61|.49|
> |gemini-3-flash|.99|.96|.83|.80|.81|.66|.63|.92|.89|.76|.75|.7|.57|.45|
> |gpt-5.4-mini-V|.99|.99|.98|.97|.95|.97|.94|.99|.99|.99|.98|.99|.95|.97|
> |gemini-3-flash-V|.99|.98|.91|.92|.91|.82|.84|.99|.99|.95|.96|.93|.9|.86|
>
> **Q1.** What factors lead to VLMs outperforming TSLMs on BEDTime?
>
> We hypothesize 3 complementary factors:
>
> - Our descriptions target structural features that, when rendered as plots, are easily detected by vision encoders, whereas TSLMs must reconstruct these from numeric arrays.
> - VLMs are trained on chart-to-text tasks, giving them proximity to ours.
> - TSLMs face instruction-following limitations stemming from their training pipelines. ChatTime starts from LLaMA-2-7B-Base, building instruction-following through fine-tuning on only 25K Alpaca samples, resulting in the high refusal rates (Appendix G, Table 16). ChatTS mitigates this by fine-tuning Qwen2.5-14B-Instruct, thereby performing competitively against comparable-sized LLMs. VLMs inherit robust instruction-following from their fully trained LLM backbones.
>
> **Q2.** What failure modes arise in generation, and do models consistently miss specific structural properties?
>
> We observe 3 failure modes:
>
> - Temporal localization is the hardest across all modalities: even gpt-4o-vision scores notably lower on position identification than on pattern recognition, indicating *where* a feature occurs is harder to describe than *what* it is.
> - LLMs hallucinate structural properties inconsistent with the input as a consequence of operating on tokenized numeric strings.
> - TSLMs exhibit low abstraction, often describing time series step-by-step rather than summarizing structure. Across all models, generation-to-ground truth entailment consistently exceeds the reverse, indicating models capture *some* structural properties correctly but fail to describe *all* properties present in the ground truth.
>
> We include these insights with extended NLI results in the revised manuscript.
>
> **L1.** Bedtime only contains univariate time series.
>
> Multivariate time series are indeed important (acknowledged in Lines 425–431). However, we are unaware of any datasets containing structural descriptions of multivariate time series and hope our work inspires such data curation. Extending to multivariate descriptions not only requires substantial data collection but also introduces unique challenges that warrant dedicated future research. Univariate time series remain extremely common and underpin many real-world applications, making contributions in this space broadly impactful.

---

> > ### Author Rebuttal · Reviewer_UCPf · 2026-04-02
> >
> > Thanks for the rebuttal, which solves my concern about the insights from experiments, especially for the failure modes. I will keep my score, which is already positive.

---

### Official Review · Reviewer_6swr · 2026-03-08

**Soundness:** 4
**Presentation:** 4
**Significance:** 2
**Originality:** 3
**Overall Recommendation:** 5
**Confidence:** 4

**Summary:**

The paper proposes a benchmark for testing AI models' capabilities of describing time series. They propose three tasks: recognition (does the proposed caption describe the given time series), differentiation (which caption is the right description for the given time series), and generation (generate a caption for the given time series). They preprocess a dataset of five existing datasets and evaluate 13 different state-of-the-art models between LLMs, vision models and specific time series foundation models.
The authors show that dedicated time series foundation models often do not perform well in time series description, that vision language models perform best, that out-of-the-box large language models perform worst, and that all methods are not really robust to noise in the data.

**Compliance With Llm Reviewing Policy:**

Affirmed.

**Final Justification:**

The paper proposes a benchmark for testing AI models' capabilities of describing time series. The authors provide experiments and evaluations on multiple models and datasets to support their claims with a well-structured presentation. While the work focuses on specific time series description capabilities, the proposed unsaturated benchmark can be part of a broader evaluation ecosystem, supporting future advancements in the field. The rebuttal addresses my questions and I confirm my positive assessment.

**Key Questions For Authors:**

- Is there a reason why most recent off-shelf large language models such as Gemini 3 Pro, GPT 5 or Claude 4.5 etc have not been considered in the evaluation? These models (e.g. Gemini 3 Pro) are thought to be much stronger in multimodal modeling. Results on more recent models would make the claims more impactful.
- What is the intra-modal rank reported in Table 2? (It may help the reader to explain it somewhere)

**Limitations:**

A few comments on limitations:
- Time series description is one useful benchmark for time series foundation models, but it only represents one perspective for evaluating their performance
- Results use proprietary models that are slightly outdated (now released almost one year ago). If more recent models already saturate it, the relevance of this work may be lower.
- Specialized time series foundation model do not perform well probably because of misalignment between training and downstream task?

**Strengths And Weaknesses:**

**Soundness**:
The paper is technically sounds and extensively justifies the claims by providing experiments and evaluations on multiple models and datasets. The evaluation methods for the three proposed tasks are appropriately justified. Furthermore, the authors propose several studies on robustness analysis.

**Presentation**:
The paper reads well and is well-structured, it is easy to read and follow: it introduces the problem, goes through the most relevant related work, describes the method and shows the experimental results.

**Significance**:
Systematic benchmarking of time series analysis capabilities of foundation models is critical for making progress in the field. However, I believe we should emphasize that qualitative time series description is only one target capability of a time series foundation model. The tasks the authors propose isolate one aspect of time series analysis, and should be complemented by additional benchmarks in the future.


**Originality**:
The work provides understanding of current performance of state-of-the-art models on time series analysis, structural understanding and description in natural language. It introduces a currently unsaturated benchmark for time series description, encouraging the field to make progress in this direction. The authors adopt an automated way of evaluating the performance on generating time series description using bidirectional entailment, which looks interesting and better-thought than LLM as judge.

---

> ### Author Rebuttal · Authors · 2026-03-31
>
> We thank the reviewer for their thoughtful and constructive feedback. We are encouraged that they find our work “technically sound and extensively justifies the claims,” describe the paper as “well-structured”, “easy to read and follow,” and recognize our open-generation evaluation approach as “interesting and better-thought than LLM as judge.” Below, we address each comment.
>
> **W1/L1.** BEDTime evaluates an important capability—qualitative description—and should be complemented by benchmarks targeting other aspects of TSLMs.
>
> We intentionally isolate description as just one foundational component of what we hope becomes a broader evaluation ecosystem for time series foundation models. We revised our introduction and related work to better situate BEDTime among existing time series–language benchmarks.
>
> **Q1/L2.** Why exclude recent models? If newer models saturate the benchmark, its relevance may be reduced.
>
> A core motivation of BEDTime is guiding development of smaller, purpose-built models for multimodal time series description—proprietary models establish upper bounds but are not the end goal. Still, we have added gpt-5.4-mini and gemini-3-flash (abbr. gpt-5.4 , gemini-3) (LLM and VLM variants) to our evaluation; Gemini-3-Pro has been deprecated and is unavailable via API, Claude 4.5 was excluded due to API access constraints. Recognition, Differentiation and Open Generation results are in Tables 1 and 2, with robustness and CoT experiments in Figures 1 and 2.
>
> **The overall findings of our paper are unchanged in light of these results.** Frontier models improve modestly over predecessors, gpt-5.4-mini gains ~11% / ~12% (LLM) and ~6% / ~5% (VLM), while gemini-3-flash gains ~10% / ~11% (LLM) and ~10% / ~6% (VLM) on Recognition / Differentiation respectively—but no model saturates the benchmark: even the best frontier VLM remains well below ceiling on real-world datasets like NICU-HR and TRUCE-Stock. VLMs continue to outperform LLMs, all models remain fragile to realistic perturbations, and CoT aids LLM performance. We incorporated these results into the revised manuscript.
>
> Table 1. Recognition (R) Differentiation (D) and Accuracy on all datasets using DTW distance for negative sampling
>
> |Model|R-TRUCE-Stock|R-TRUCE-Synthetic|R-SUSHI|R-TaxoSynth|R-NICU-HR|D-TRUCE-Stock|D-TRUCE-Synthetic|D-SUSHI|D-TaxoSynth|D-NICU-HR|
> |-|-|-|-|-|-|-|-|-|-|-|
> |gpt-5.4|0.68|0.86|0.89|0.81|0.63|0.61|0.84|0.76|0.76|0.96|
> |gemini-3|0.65|0.79|0.81|0.76|0.62|0.58|0.75|0.72|0.68|0.95|
> |gpt-5.4-V|0.79|0.88|0.97|0.9|0.69|0.73|0.88|0.98|0.84|0.98|
> |gemini-3-V|0.73|0.84|0.91|0.87|0.66|0.68|0.83|0.94|0.77|0.96|
>
> Table 2. NLI entailment (%) on SUSHI (Gen = Generation; GT = Ground Truth)
>
> |Model|Bi-directional|Gen → GT|GT → Gen|
> |-|-|-|-|
> |gpt-5.4-V|16.29|58|31.14|
> |gemini-3-V|14.14|42.43|14.14|
> |gpt-5.4|5.43|35.21|8.86|
> |gemini-3|2.79|7.93|7.01|
>
> Figure 1:​​ https://bit.ly/4sCsBMZ
>
> Figure 2: https://bit.ly/4lYftiR
>
> **Q2.** What is the intra-modal rank reported in Table 2?
>
> The intra-modal rank averages each model's within-modality accuracy rank across datasets. Following Q1/L2, we updated rankings to include gpt-5.4-mini and gemini-3-flash. We clarified this definition in the revised manuscript and include the table below.
>
> Table 3. Intra-modal ranks for all datasets and models for Recognition (R) and Differentiation (D)
>
> |Model|R-TRUCE-Stock|R-TRUCE-Synthetic|R-SUSHI|R-TaxoSynth|R-NICU-HR|R-Rank|D-TRUCE-Stock|D-TRUCE-Synthetic|D-SUSHI|D-TaxoSynth|D-NICU-HR|D-Rank|Overall Rank|
> |-|-|-|-|-|-|-|-|-|-|-|-|-|-|
> |gpt-5.4|1|1|1|1|1|1| 1|1|1|1|1|1|1|
> |gemini-3| 3|4|3|3|2|3|2|3|2|4|2|2.6|2.8|
> |gpt-4o+LT|2|2|2|2|3|2.2|3|2|3|2|2|2.4|2.3|
> |gpt-4o|4|3|4|4|3|3.6|5|4|4|3|3|3.8|3.7|
> |gemini-2.0|6|5|5|5|4|5|4|6|5|5|3|4.6|4.8|
> |qwen2.5-14b|5|5|6|6|5|5.4|4|5|6|6|4|5|5.2|
> |qwen2.5-7b|3|6|6|7|6|5.6|6|7|8|8|5|6.8|6.2|
> |phi-3.5|7|8|7|7|7|7.2|7|8|7|7|6|7|7.1|
> |llama-3.1-8b|8|7|8|8|8|7.8|8|9|9|9|7|8.4|8.1|
> |gpt-5.4-V|1|1|1|1|1|1|1|1|1|1|1|1|1|
> |gemini-3-V|3|2|3|3|2|2.6|2|2|3|3|2|2.4|2.5|
> |gpt-4o|2|3|2|2|3|2.4|3|3|2|2|2|2.4|2.4|
> |gemini-2.0|5|4|4|4|3|4|4|4|4|5|3|4|4|
> |qwen2.5-vl-7b|6|4|6|5|4|5|5|5|5|4|4|4.6|4.8|
> |phi-3.5-vision|4|5|5|6|5|5|6|6|6|6|5|5.8|5.4|
> |chatts|1|1|1|1|1|1|1|1|1|1|1|1|1|
> |chattime|2|2|2|2|1|1.8|2|2|2|2|2|2|1.9|
>
> **L3.** Does TSLM underperformance stem from misalignment between their training objectives and BEDTime's tasks?
>
> Both TSLMs are trained for reasoning and MCQ-style question answering, so their objectives are aligned with BEDTime's tasks. The underperformance stems from degraded instruction-following. ChatTime builds from LLaMA-2-7B-Base (non-instruction-tuned) with fine-tuning only 25K Alpaca samples for instruction-following, producing the high refusal rates in Appendix G (Table 16). ChatTS (built on Qwen2.5-14B-Instruct) performs better—competing with comparably-sized LLMs (main manuscript, Table 2). Both consistently fall short of VLMs, which benefit from visual encoders that make structural properties directly perceptible.

---

> > ### Author Rebuttal · Reviewer_6swr · 2026-04-02
> >
> > Thank you for the detailed reply and for showing results with updated state of the art models. It is interesting to see that performance improved but the dataset has not yet been saturated. Regarding Gemini 3 Pro, I believe it has been replaced by Gemini 3.1 Pro. If a core motivation for this work is guiding the development of smaller multimodal models for time series, I believe it would make sense to mention it more clearly in the main text. The rebuttal addresses my questions and I confirm my positive assessment.

---

### Official Review · Reviewer_T7xk · 2026-03-13

**Soundness:** 3
**Presentation:** 3
**Significance:** 3
**Originality:** 2
**Overall Recommendation:** 4
**Confidence:** 2

**Summary:**

The paper introduces BEDTime, a unified benchmark for evaluating the ability of multi-modal models to describe structural properties of univariate time series through natural language. The authors argue that recognizing and generating descriptions of features like trends, seasonality, and anomalies is a foundational capability for time series reasoning. BEDTime unifies five existing datasets resulting in over 46,000 time series-description pairs, primarily from real-world sources. The benchmark evaluates three tasks: description recognition, description differentiation, and open-ended generation.

**Compliance With Llm Reviewing Policy:**

Affirmed.

**Final Justification:**

Having read the rebuttal, I find that the authors have satisfactorily addressed most of my concerns around experiments and results. This has helped clarify the contribution, and I will maintain my positive score.

**Key Questions For Authors:**

- Could the experiments include additional examples involving extremely short and extremely long time series? The models struggle substantially on the TRUCE-Stock dataset, which may be due to the inherent noise in financial data or the short sequence lengths.
- Models from different modalities receive different information density from the input time series. Could this discrepancy also affect their performance?
- For the human evaluation, how were the 340 samples distributed across models and subclasses, and were annotators shown the original time series plots or only the generated and reference descriptions?

**Limitations:**

yes

**Strengths And Weaknesses:**

Strengths:
- The evaluation data are derived from real-world sources, allowing the benchmark to assess models’ reasoning and application capabilities under realistic demands.
- The benchmark aggregates five public datasets and reformats them under a unified protocol, resulting in 46,843 pairs with substantial variation in sequence length, description style, and data source, thereby enabling evaluation from multiple complementary perspectives.
- The authors combine automated Natural Language Inference and GPT-5-based judging with a six-criterion human expert assessment to provide a more holistic evaluation of generated text. They also conduct a human evaluation to assess whether the generated descriptions are generic, accurate, interpretable, and fluent, which improves the credibility of the evaluation.
- The paper provides detailed evaluation and analysis with respect to different time-series characteristics, which is useful for subsequent research on model architecture design and training strategies.

Weaknesses：
- This current benchmark focuses exclusively on univariate time series, which may not capture the complexity of cross-channel dependencies found in multivariate real-world applications.
- While the human evaluation is detailed, it is conducted only on a subset of the SUSHI dataset (340 samples), which might not fully represent the linguistic nuances of the other four datasets.
- The comparison across model families is somewhat uneven in terms of interface. LLMs receive comma-separated values, VLMs receive plots, and TSLMs receive numeric vectors directly. This is part of the intended multimodal comparison, but it also makes it hard to disentangle model capability from input representation advantage. A more controlled representation ablation would clarify this issue.
- The paper shows that CoT improves LLM performance, but it does not deeply analyze when CoT helps and whether those gains come from better temporal reasoning or simply more careful answer formatting. Additional error analysis would improve the interpretability of this finding.

---

> ### Author Rebuttal · Authors · 2026-03-31
>
> We thank the reviewer for the constructive feedback. We appreciate recognition of our real-world evaluation data for assessing models "under realistic demands" and of our analysis as "useful for subsequent research on model architecture design and training strategies." We address each concern below.
>
> **W1.** BEDTime is limited to univariate time series.
>
> Multivariate time series are indeed important (acknowledged in Lines 425–431). However, we are unaware of any datasets containing structural descriptions of multivariate time series and hope our work inspires such data curation. Extending to multivariate descriptions requires not only substantial data collection but also introduces enough unique challenges that warrant dedicated future research. Univariate time series remain extremely common and underpin many real-world applications, making contributions in this space broadly impactful.
>
> **W2.** Scale of the human evaluation.
>
> Full-dataset human evaluation is prohibitive: ~1 minute per description, 3 annotators spent ~21 hours evaluating 140 descriptions each for 3 models (340 total). Given this constraint, we chose SUSHI as its time series and descriptions are the longest and most complex, making it the strongest single test case. To ensure coverage, sampling was stratified across all 7 SUSHI classes (20 per class), with 40 shared across annotators for inter-annotator agreement. We have added these details to Section 3.1 of our manuscript and expanded our NLI experiments.
>
> **W3/Q2.** Uneven input representations across models.
>
> We agree different modalities may include different amounts of information about each time series. However, this does not impact our results because our experiments are designed to test this. Each model receives input in its native modality, following the TSLM literature [1, 2, 3]. These modalities include sufficient resolution to describe each time series, so differences in performance are not due to some modalities being too low quality. If any exception exists, it’s vision as it sacrifices the most due to lack of numeric precision. But VLMs are consistently the best-performing models even when controlling for the base LLM (Figure 5). Therefore, to the best of our knowledge our results are not biased towards an input modality. We have added this discussion to our Limitations section.
>
> **W4.** The source of the performance gains on LLMs from CoT is not analyzed.
>
> We analyzed CoT's effects across all model–dataset pairs for Recognition (R) and Differentiation (D). We derive the following findings:
> -  CoT's overall effect is modest: mean absolute accuracy increase is +0.03 across all model–dataset pairs for both tasks. Focusing on the two models showing gains exceeding 5%, we identify the sources of improvement:
> - Largest relative gains come from reduced refusal rates: Qwen-7b improves by +11.85% (R) and +6.58% (D) on average, with the highest gains on SUSHI, where refusal rates drop from 24.7% to 11.8% (R) and 27.2% to 19.4% (D).
> - GPT-4o shows gains on TRUCE-Stock where CoT reduces systematic positional bias toward the first available option: 18.1% (R) and 11.7% (D).
>
> We have added this analysis to Section 4.2.3.
>
> **Q1.** Would testing on extreme sequence lengths clarify model performance, especially for TRUCE-Stock?
>
> First, we completely agree that sequence length likely drives description ability. This is why BEDTime includes datasets with a wide range of lengths (12 to 2048 time steps) and we experiment with varying sequence lengths (Section 4.2.3).
>
> For TRUCE-Stock results specifically, the lower performance appears primarily driven by high noise in the data, not sequence length: TRUCE-Stock and TRUCE-Synthetic share the same sequence length (12) and similar description lengths and source (Table 1). The key difference is time series source. TRUCE-Stock contains real financial data from Google Finance, which is inherently noisy at short horizons, while TRUCE-Synthetic contains cleaner synthetic signals. We have clarified this distinction in the revised manuscript.
>
> **Q3.** What was the sampling and annotation setup in human evaluation?
>
> Annotators were shown matplotlib-rendered plots, raw time series values, and model-generated descriptions from each of the 3 evaluated models (Table 3), alongside detailed definitions of each of the 6 binary criteria (Section 4.2.2). We used stratified random sampling across all 7 SUSHI classes (20 samples per class), yielding 140 descriptions per annotator—40 shared for inter-annotator agreement, 100 unique to each, totaling 340 evaluated descriptions. We have added this description in the revised manuscript.
>
> **References:**
>
> [1] Xie et al., 2025. “ChatTS: Aligning Time Series with LLMs via Synthetic Data for Enhanced Understanding and Reasoning”
>
> [2] Cai et al., 2024. "TimeSeriesExam: A Time Series Understanding Exam"
>
> [3] Zhou et al., 2025. "CaTS-Bench: Can Language Models Describe Numeric Time Series?"

---

> > ### Author Rebuttal · Reviewer_T7xk · 2026-04-03
> >
> > Thank the authors for the detailed rebuttal. The additional experimental details and methodological clarifications have mostly addressed the concerns I raised.
> > I will not change my score, as it is already positive.

---

### Official Review · Reviewer_FvbP · 2026-03-13

**Soundness:** 3
**Presentation:** 3
**Significance:** 3
**Originality:** 3
**Overall Recommendation:** 5
**Confidence:** 4

**Summary:**

This paper proposes a unified benchmark for time series description named BEDTime. The authors reformat five public datasets into this unified evaluation setting and conduct a systematic study of 13 state-of-the-art models on the three proposed tasks. The experiments combine automatic metrics with human evaluation and further examine model robustness under several types of perturbations. The results show that VLM perform best on this benchmark, pure LLM perform worst, and even methods specifically designed for time series do not have clear advantages. Based on these findings, the paper argues that current multimodal time series language models still exhibit substantial limitations in more basic structural description abilities, while BEDTime provides a unified benchmark framework for future evaluation in this area.

**Compliance With Llm Reviewing Policy:**

Affirmed.

**Final Justification:**

The primary concerns I raised have been fully resolved in the authors' responses.

**Key Questions For Authors:**

1. The paper evaluates 13 advanced models in total, but only 2 of them are TSLMs. Since the number of TSLMs is quite small and the coverage across model groups is not balanced, could this affect the results and make some conclusions less reliable?
2. In the generation task, the paper only evaluates the single best method from each category from the first two tasks. Why did authors choose this setting? Would this limit the evaluation of the generation task? I suggest either evaluating more models on the generation task to obtain more reliable results or explaining more clearly why this sampling choice is reasonable. If the authors believe that methods performing well on task1 and task2 are also representative for task 3, could they provide some theoretical or empirical analysis to explain the connection between these tasks?
3. Different model categories in the paper use different input modalities. How do the authors rule out the effect of differences in input modality and input quality on the experimental results? Could these differences affect a fair evaluation of the models’ own capabilities?
4. As the main contribution of this paper lies in evaluation rather than method/algorithm design, it would better to make the code and data publicly available, at least to the reviewers/ACs/SACs, to fully evaluate the significance of the paper.

**Limitations:**

yes

**Strengths And Weaknesses:**

Strengths
- The problem of designing a unified benchmark to evaluate models’ ability to describe the structural properties of time series is highly necessary, which not only helps reassess prior work but also provides a clearer direction for future study.
- The paper introduces three tasks (i.e., identifying the correct description for a given time series, distinguishing correct descriptions from incorrect ones, and generating natural-language descriptions that capture the key structural properties of the input series) and conducts a broad empirical study over 13 advanced models on 5 representative reformatted datasets, together with robustness tests. Technically, the three tasks are complementary, which makes the benchmark design reasonably sound. As for datasets, they include real-world time series with diverse structural patterns, which strengthens the reliability of the evaluation.
- The paper is overall well structured and easy to follow. It clearly explains both the connections to and differences from related work.


Weaknesses
- The model selection in BEDTime is kind of unbalanced across LLMs, VLMs, and TSLMs, with only two available TSLMs included, which may bias the conclusions and make some category-level claims overgeneralized.
- Only the best performing model from each category in the first two tasks is further evaluated for generation tasks, resulting in limited evaluation.
- The experimental setup is described too briefly, which may make reproduction more difficult, and it would benefit from adding necessary details.
- The reproducibility is somewhat poor. Code and data, which are highly important for evaluating the significance of a evaluation and benchmark paper, are not available.

---

> ### Author Rebuttal · Authors · 2026-03-31
>
> We thank the reviewer for their thoughtful and constructive feedback. We are encouraged that the reviewer finds our work "highly necessary" noting that the benchmark "helps reassess prior work" and provides "a clearer direction for future study." Below, we address the concerns.
>
> **W1/Q1.** Results include only 2 TSLMs compared to 7 LLMs and 4 VLMs.
>
> We thank the reviewer for suggesting we clarify our TSLM selection criteria. Time series–language modeling is still a new and emerging area, and as documented in Table 5 (Appendix A), many “TSLMs” output time series, not natural language, placing them outside BEDTime's evaluation scope. So overall, the chosen TSLMs remain representative of publicly available models that support our problem setting. We agree that adding more TSLMs as they come out will be valuable and BEDTime (fully public) is designed to support evaluation of future models in this setting.
>
> **W2/Q2.** The generation task focuses on the top-performers from each category.
>
> We note that there are two evaluations for generation: human evaluations and automated NLI. The human evaluations, are bottlenecked by annotation time so we had to be strategic—annotators spent **1 minute on average per time series (3 model responses * 140 time series * 3 annotators = ~1260 minutes total)**. Following the reviewer's suggestion, we expanded the automated NLI evaluation to all models (Table 1). This supports our core finding, and  preserves the same modality hierarchy observed in Recognition and Differentiation. To validate this we also computed rank correlations between mean Recognition/Differentiation ranks and Generation ranks across all models: Spearman ρ = .856/.830 and Kendall τ = .700/.661. We have included all new results in the revised manuscript.
>
> Table 1. NLI entailment (%) on SUSHI. (Gen = Generation, GT = Ground Truth)
>
> |Model|Bi-directional|Gen → GT|GT → Gen|
> |-|-|-|-|
> |gemini-2.0-V|2.79|8|5.14|
> |gemini-2.0|2.21|15.43|2.43|
> |qwen2.5-vl-7b|2.07|14.15|2.21|
> |qwen2.5-14b|1.93|7.21|2.07|
> |Phi-3.5-vision|1.64|8.36|2|
> |qwen2.5-7b|1.5|6.21|1.79|
> |phi-3.5|1.21|1.57|1.5|
> |chattime-7b|1.08|2.21|1.29|
> |llama-3.1-8b|.07|1|.07|
>
> **W3/W4/Q4.** Code/data availability and additional experimental setup details.
>
> We thank the reviewer for suggesting we clarify some of our experimental details and code availability in the paper.
>
> First, we’re glad to say that **all code and data are already available**—see the supplementary .zip file we submitted via OpenReview and the mentions in the Abstract (Lines 53-54) and Contribution 3 in our introduction (Lines 92-93). As stated in the paper, all data and code will be released publicly via Hugging Face and GitHub.
>
> Second, *we have moved experimental details from our Appendices (B, C, and F) into Section 3*. To further improve reproducibility, we have added an end-to-end walkthrough of the evaluation pipeline in the Appendix, covering data loading through evaluation, to help future users replicate our results or integrate new datasets and models into BEDTime.
>
> **Q3.** Do different input representations across modalities affect fair comparison of model capabilities?
>
> We agree different modalities may include different amounts of information about each time series. However, this does not impact our results because our experiments are designed to test this. Each evaluated model receives input in its native modality, following the TSLM literature [1, 2, 3]: comma-separated text for LLMs, line graphs for VLMs, and raw numeric vectors for TSLMs. These modalities include sufficient resolution to describe each time series, so differences in performance are not due to some modalities being too low quality.  If there were a case to be made against any single modality, it would be against vision, which sacrifices the most numeric precision. However, we find the vision models are consistently the best-performing models. This is true even when controlling for the base model (Figure 5). Based on all of these findings, to the best of our knowledge our results are not biased towards an input modality. We have added this discussion to our Limitations section.
>
> **References:**
>
> [1] Xie et al., 2025. “ChatTS: Aligning Time Series with LLMs via Synthetic Data for Enhanced Understanding and Reasoning”
>
> [2] Cai et al., 2024. "TimeSeriesExam: A Time Series Understanding Exam"
>
> [3] Zhou et al., 2025. "CaTS-Bench: Can Language Models Describe Numeric Time Series?"

---

> > ### Author Rebuttal · Reviewer_FvbP · 2026-04-04
> >
> > Thank the explanations and additonal work from the authors. My concerns have been fully resolved, so I would like to raise my rating.

---

### Decision · Program_Chairs · 2026-04-30

**Decision:**

Accept (regular)

**Comment:**

Reviewers across the board found the benchmark highly necessary and technically sound, praising the inclusion of real-world datasets, the holistic evaluation combining automated NLI and human assessment, and the systematic robustness analysis. The authors successfully addressed initial concerns during the rebuttal by clarifying selection criteria for specialized models, providing qualitative error analyses, and expanding experiments to include the latest frontier models. Given the strong consensus among reviewers and the paper’s contribution to an important but underserved area of AI evaluation, this paper is recommended for acceptance.